# Molecule Generation by Principal Subgraph Mining and Assembling

**Xiangzhe Kong**[1]  **Wenbing Huang**[4,5]*  **Zhixing Tan** [1]  **Yang Liu**[1,2,3]*
[1]Dept. of Comp. Sci. & Tech., Institute for AI, BNRist Center, Tsinghua University
[2]Institute for AIR, Tsinghua University  [3]Beijing Academy of Artificial Intelligence
[4]Gaoling School of Artificial Intelligence, Renmin University of China
[5] Beijing Key Laboratory of Big Data Management and Analysis Methods, Beijing, China
Jackie_KXZ@outlook.com, hwenbing@126.com, {zxtan, liuyang2011}@tsinghua.edu.cn

## Abstract

Molecule generation is central to a variety of applications. Current attention has been paid to approaching the generation task as subgraph prediction and assembling. Nevertheless, these methods usually rely on hand-crafted or external subgraph construction, and the subgraph assembling depends solely on local arrangement. In this paper, we define a novel notion, *principal subgraph* that is closely related to the informative pattern within molecules. Interestingly, our proposed merge-and-update subgraph extraction method can automatically discover frequent principal subgraphs from the dataset, while previous methods are incapable of. Moreover, we develop a two-step subgraph assembling strategy, which first predicts a set of subgraphs in a sequence-wise manner and then assembles all generated subgraphs globally as the final output molecule. Built upon graph variational auto-encoder, our model is demonstrated to be effective in terms of several evaluation metrics and efficiency, compared with state-of-the-art methods on distribution learning and (constrained) property optimization tasks.

## 1 Introduction

Generating chemically valid molecules with desired properties is central to a variety of applications in drug discovery and material science. In contrast to searching countless potential candidates by expert chemists or pharmacologists, designing generative models that can produce molecules automatically is more efficient and now a prevailing topic in machine learning. Recently, there have been various works[18, 26, 47, 24, 9, 39, 20] utilizing graphs to characterize the distribution of molecules.

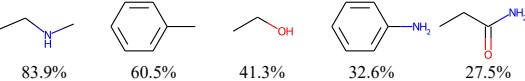

83.9%    60.5%    41.3%    32.6%    27.5%

Figure 1: Five frequent subgraphs in ZINC250K [16, 23] marked with their frequencies of occurrence.

In general, these graph-based generation methods can be clustered into two categories: the atom level [25, 47, 26, 20] and the subgraph level [18, 19, 46], in terms of what the basic generation unit is. Compared with the atom-level methods, exploring subgraphs for generating molecules (as illustrated in Figure 1) exhibits three potential benefits. First, it can capture the regularities of atomic combinations and discover repetitive patterns in molecules, thus more capable of generating realistic molecules. Second, it is able to reflect chemical properties, since it has been revealed that the chemical properties are closely related to certain substructures (i.e., subgraphs) [31, 20]. Finally, leveraging subgraphs as the building blocks enables efficient training and inference, as the searching time is remarkably reduced by representing a molecule as a set of subgraphs other than atoms.

---

*Wenbing Huang and Yang Liu are corresponding authors.

36th Conference on Neural Information Processing Systems (NeurIPS 2022).

However, current subgraph-level methods [18, 19, 46] are still suboptimal in two senses. For one thing, the vocabulary of the molecular fragments is constructed by simple hand-crafted rules [18, 19] or borrowed straightly from external chemical fragment libraries [46], which is unable (at least uncertain) to reveal the frequent patterns existing in the dataset. For another, the prediction and the assembling of the subgraphs are conducted either autoregressively [19, 46] or according to a pre-defined tree structure [18], both of which are inflexible and defective, since each newly predicted subgraph is only allowed to attach a local set of previous subgraphs during the assembling process.

To address the above two pitfalls, this paper proposes a novel framework: Principal Subgraph Variational Auto-Encoder (PS-VAE). PS-VAE first creates a vocabulary of molecular fragments from any given dataset, which starts from all distinct atoms and then merges the neighboring fragments as a new one to update the vocabulary. Interestingly, the fragments derived by such a merge-and-update strategy are actually *principal subgraphs* which, as a novel concept defined by this paper, represent the frequent and largest repetitive patterns of molecules. We also theoretically prove that any principal subgraph can be universally covered by our vocabulary. Moreover, we develop a two-step subgraph assembling strategy, which first sequentially predicts a set of fragments and then assembles all generated subgraphs globally. This two-step fashion makes our method less permutation-dependent and focus more on global connectivity against traditional methods [18, 19, 46].

We conduct extensive experiments on the ZINC250K [16] and QM9 [6, 37] datasets. Results demonstrate that our PS-VAE outperforms state-of-the-art models on distribution learning, (constrained) property optimization as well as GuacaMol goal-directed benchmarks [7]. Besides, PS-VAE is efficient and about six times faster than the fastest autoregressive baseline.

## 2 Related Work

**Molecule Generation.** Regarding the representation of molecules, existing generation models can be divided into two categories: text-based and graph-based methods. Text-based models [13, 23, 5] usually adopt the Simplified Molecular-Input Line-entry System ([40], SMILES) to describe each molecule, which is simple and efficient. However, they are not robust because slight perturbations in SMILES could result in significant changes in molecule structure [47, 24]. The graph-based counterparts [9, 39, 20], therefore, have gained increasing attention recently. Li et al. [25] proposed a generation model for graphs and demonstrated it performed better than the text-based strategy. You et al. [47] used reinforcement learning to generate molecules sequentially under the guidance of mixed rewards in terms of the chemical validity and other property scores. Popova et al. [34] proposed MRNN to autoregressively generate nodes and edges. Shi et al. [39] designed a flow-based autoregressive model and exploited reinforcement learning for goal-directed generation. However, these models use atoms as the basic generation unit, leading to long sequences and therefore time-consuming training process. On the contrary, our method applies subgraph-level representation, which not only captures chemical properties but also is computationally efficient.

**Subgraph-Level Molecule Generation** Several works have been developed for subgraph-level generation. In particular, Jin et al. [18] suggested generating molecules in the form of junction trees where each node is a ring or edge. Jin et al. [19] decomposed molecules into subgraphs by breaking all the bridge bonds. It used a complex hierarchical model for polymer generation and graph-to-graph translation. Yang et al. [46] integrated reinforcement learning with the subgraph vocabulary created from existing chemical fragment libraries. Different from Jin et al. [18, 19], Yang et al. [46] which use manual rules to extract subgraphs or utilize existing libraries, we automatically extract frequent principal subgraphs to better capture the regularities in molecules for subgraph-level decomposition. Our work also relates to traditional subgraph mining literature [15, 45, 33]. While seeking generic frequent subgraphs is known to be NP-hard [22, 17], our work focuses on finding frequent principal subgraphs by merging two adjacent fragments, which alleviates time complexity to a large extent. To the best of our knowledge, we are the first to utilize frequent subgraphs in generation tasks.

## 3 Our Proposed PS-VAE

This section presents the details of the proposed PS-VAE, consisting of the principal subgraph definition and extraction in Section 3.1 and the two-step generation in Section 3.2.

## 3.1 Principal Subgraph

A molecule can be represented as a graph $\mathcal{G} = \langle \mathcal{V}, \mathcal{E} \rangle$, where $\mathcal{V}$ is a set of nodes corresponding to atoms and $\mathcal{E}$ is a set of edges corresponding to chemical bonds. We define a *subgraph* of $\mathcal{G}$ as $\mathcal{S} = \langle \tilde{\mathcal{V}}, \tilde{\mathcal{E}} \rangle \subseteq \mathcal{G}$, were $\tilde{\mathcal{V}} \subseteq \mathcal{V}$ and $\tilde{\mathcal{E}} \subseteq \mathcal{E}$. We say subgraph $\mathcal{S}$ *spatially intersects* with subgraph $\mathcal{S}'$ if there are certain atoms in a molecule belong to both $\mathcal{S}$ and $\mathcal{S}'$, denoted as $\mathcal{S} \cap \mathcal{S}' \neq \emptyset$. Note that if two subgraphs look the same (with the same topology), but they are constructed by different atom instances, they are not spatial intersected. Similarly, if two subgraphs $\mathcal{S}$ and $\mathcal{S}'$ appear in the same molecule, we call their *spatially union* subgraph as $\mathcal{U} = \mathcal{S} \bigcup \mathcal{S}'$, where the nodes of $\mathcal{U}$ are the union set of $\tilde{\mathcal{V}}$ and $\tilde{\mathcal{V}}'$, and its edges are the union of $\tilde{\mathcal{E}}$ and $\tilde{\mathcal{E}}'$ plus all edges connecting $\mathcal{S}$ and $\mathcal{S}'$. A *decomposition* of a molecule $\mathcal{G}$ is derived as a set of non-overlapped subgraphs $\{\mathcal{S}_i\}_i^n$ and the edges connecting them $\{\mathcal{E}_{ij}\}_{i,j}^{n,n}$, if $\mathcal{G} = (\bigcup_i^n \mathcal{S}_i) \bigcup (\bigcup_{i,j}^{n,n} \mathcal{E}_{ij})$ and $\mathcal{S}_i \cap \mathcal{S}_j = \emptyset$ for any $i \neq j$. The *frequency* of a subgraph occurring in all molecules of a given dataset measures its repeability and epidemicity, which should be an important property. Formally, we define the frequency of a subgraph $\mathcal{S}$ as $c(\mathcal{S}) = \sum_i c(\mathcal{S}|\mathcal{G}_i)$ where $c(\mathcal{S}|\mathcal{G}_i)$ computes the occurrence of $\mathcal{S}$ in a molecule $\mathcal{G}_i$. Without loss of generality, we assume all molecules and subgraphs we discuss are connected.

With the aforementioned notations, we propose a novel and central concept below.

**Definition 3.1** (Principal Subgraph). We call subgraph $\mathcal{S}$ a principal subgraph, if any other subgraph $\mathcal{S}'$ that spatially intersects with $\mathcal{S}$ in a certain molecule satisfies either $\mathcal{S}' \subseteq \mathcal{S}$ or $c(\mathcal{S}') \leq c(\mathcal{S})$.

Amongst all subgraphs of the larger frequency, a principal subgraph basically represents the "largest" repetitive pattern in size within the data. It is desirable to leverage patterns of this kind as the building blocks for molecule generation since those subgraphs with a larger size than them are less frequent/reusable. We will prove that our designed vocabulary is capable of mining these subgraphs.

Now we introduce the proposed vocabulary construction process. We call each subgraph of the constructed vocabulary as a *fragment*. We generate all fragments via the following stages:

**Initialization**. The vocabulary $\mathbb{V}$ is initialized with all unique atoms (subgraph with one node).

**Merge**. For every two neighboring fragments $\mathcal{F}$ and $\mathcal{F}'$ in the current vocabulary, we merge them by deriving the union $\mathcal{F} \bigcup \mathcal{F}'$. Here, the neighboring fragments of a given fragment $\mathcal{F}$ in a molecule is defined as the ones that contain at least one first-order neighbor nodes of a certain node in $\mathcal{F}$.

**Update**. We count the frequency of each identical merged subgraph in the last stage. We choose the most frequent one as a new fragment in the vocabulary $\mathbb{V}$. Then, we go back to the merge stage until the vocabulary size reaches the predefined number N.

---

**Algorithm 1** Principal Subgraph Extraction

1: **Input:** A set of graphs $\mathcal{D}$ and the desired number $N$ of principal subgraphs to learn.
2: **Output:** A set of principal subgraphs $\mathbb{V}$ and the counter $\mathcal{C}$ of principal subgraphs.
3: $\mathbb{V} \leftarrow \{\text{GraphToSMILES}(\langle \{a\}, \emptyset \rangle)\}$; {*Initially, $\mathbb{V}$ corresponds to all atoms $a$ that appear in $\mathcal{D}$.*}
4: $N' \leftarrow \max(N, |\mathbb{V}|)$;
5: **while** $|\mathbb{V}| < N'$ **do**
6:     $\mathcal{C} \leftarrow \text{EmptyMap}()$; {*Initialize a counter.*}
7:     **for** $\mathcal{G}$ in $\mathcal{D}$ **do**
8:         **for** $\langle \mathcal{F}_i, \mathcal{F}_j, \mathcal{E}_{ij} \rangle$ in $\mathcal{G}$ **do**
9:             $\mathcal{F} \leftarrow \text{Merge}(\langle \mathcal{F}_i, \mathcal{F}_j, \mathcal{E}_{ij} \rangle)$; {*Merge neighboring fragments into a new fragment.*}
10:             $s \leftarrow \text{GraphToSMILES}(\mathcal{F})$; {*Convert the graph to SMILES representation.*}
11:             $\mathcal{C}[s] = \mathcal{C}[s] + 1$; {*Update the counter. The initial value is 0.*}
12:         **end for**
13:     **end for**
14:     $s = \text{TopElem}(\mathcal{C})$; {*Find the most frequent merged fragment.*}
15:     $\mathcal{F} \leftarrow \text{SMILESToGraph}(s)$; {*Convert the SMILES string to graph representation.*}
16:     $\mathbb{V} \leftarrow \mathbb{V} \cup \{s\}$; $\mathcal{D}' \leftarrow \{\}$;
17:     **for** $\mathcal{G}$ in $\mathcal{D}$ **do**
18:         $\mathcal{G}' \leftarrow \text{MergeSubGraph}(\mathcal{G}, \mathcal{F})$; {*Update the graph representation if possible.*}
19:         $\mathcal{D}' \leftarrow \mathcal{D}' \cup \{\mathcal{G}'\}$;
20:     **end for**
21:     $\mathcal{D} \leftarrow \mathcal{D}'$;
22: **end while**

---

(a) Initialization      (b) Iteration 1      (c) Iteration 2

Figure 2: Fragment extraction on {C=CC=C,CC=CC,C=CCC}. (a) Initialize vocabulary with atoms. (b) Fragment CC is the most frequent and added to the vocabulary. All CC are merged and highlighted in red. (c) Fragment C=CC is the most frequent and added to the vocabulary. All C=CC are merged and highlighted in green (molecules 1 and 3). After 2 iterations the vocabulary is {C, CC, C=CC}.

Figure 2 illustrates the above processes and Algorithm 1 gives the flowchart. Notice that Algorithm 1 has exploited the conversion from molecular subgraph to SMILES [40] to not only ensure the

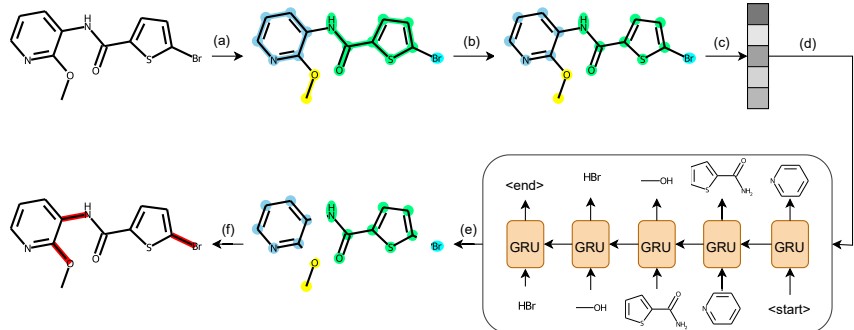

Figure 3: Overview of the *principal subgraph variational auto-encoder*. (a) Subgraph-level decomposition. Atoms and bonds of different principal subgraphs are highlighted in different colors. (b) Molecular graph. We inject subgraph-level information into the molecular graph through atom features. (c) Latent space encoding. We obtain the latent variable $z$ through the graph encoder. (d) Subgraph-level sequence generation. A sequence of principal subgraphs is autoregressively decoded from the latent variable by a GRU. (e) Incomplete molecular graph. The generated principal subgraphs form an incomplete molecular graph where inter-subgraph bonds are absent. (f) Bond completion. Completion of inter-subgraph bonds is formalized as a link prediction task for a GNN. After training, we can directly sample from the latent space to generate molecules.

uniqueness but also better calculate the occurrence of each merged subgraph by string comparison. One notable merit is that we always track the decomposition of each molecule at each iteration, which is crucial for the training of the two-step decoder introduced in the next subsection.

More importantly, we have the following theorem to demonstrate the benefit of Algorithm 1.

**Theorem 3.2.** *The vocabulary $\mathbb{V}$ constructed by Algorithm 1 exhibits the following advantages.*

(i) **Monotonicity**: *The frequency of the non-single-atom fragments in $\mathbb{V}$ decreases monotonically, namely, $\forall \mathcal{F}_i, \mathcal{F}_j \in \mathbb{V}, c(\mathcal{F}_i) \leq c(\mathcal{F}_j)$, if $i \geq j$.*

(ii) **Significance**: *Each fragment $\mathcal{F}$ in $\mathbb{V}$ is a principal subgraph.*

(iii) **Completeness**: *For any principal subgraph $\mathcal{S}$ arising in the dataset, there always exists a fragment $\mathcal{F}$ in $\mathbb{V}$ satisfying $\mathcal{S} \subseteq \mathcal{F}$, $c(\mathcal{S}) = c(\mathcal{F})$, when $\mathbb{V}$ has collected all fragments with frequency no less than $c(\mathcal{S})$.*

The conclusion by Theorem 3.2 is interesting and valuable. It states that our algorithm is able to discover frequent principal subgraphs, and any principal subgraph can be represented (at least contained) by our certain extracted fragment if the size of the vocabulary is sufficiently large. This is also a meaningful extension to traditional subgraph mining literature [17], in which seeking the most frequent subgraphs is known to be NP-hard. Algorithm 1 can somehow address this problem efficiently by focusing mainly on finding the most frequent principal subgraphs.

## 3.2 Two-step Molecule Generation

The molecule generation is modeled as a two-step task: first predicting which fragment should be chosen from the vocabulary created in the last subsection, and then proceeding with how to assemble the predicted fragments. Although in the first step we predict the fragments in a sequence-wise manner, we globally investigate if every two predicted fragments should be linked in the second step. This two-step fashion makes our method not only less permutation-dependent and thus more sample-efficient than the generic autoregressive counterparts [25], but also more flexible than the methods [18] that require predefined junction tree or adjacent connections between subgraphs. We will provide experimental evaluations in Table 4 to justify the benefit of our proposed two-step generation strategy. Below, we provide the details of the two steps.

**What to assemble.** This step is of the VAE style [21]. As shown in Figure 3, we use a GNN to encode a molecular graph $\mathcal{G}$ into a latent variable $z$. Each node $v$ has a feature vector $x_v$ which is the concatenation of three learnable embeddings: its atomic type, fragment type and the generation order of the fragment. Each edge has a feature vector indicating its bond type. Specifically, we utilize GIN with edge feature [14] as the backbone GNN network, which computes the $k$-th layer as follows:

$$a_v^{(k)} = \sum_{u \in \mathcal{N}(v)} \text{ReLU}(h_u^{(k-1)} + e_{uv}), \qquad (1)$$

$$h_v^{(k)} = h_\Theta((1 + \varepsilon)h_v^{(k-1)} + a_v^{(k)}), \qquad (2)$$

where, $h_v^{(k)}$ is the hidden activation of node $v$ ($h_v^{(0)} = x_v$), $e_{uv}$ is the edge feature between node $u$ and $v$, $\mathcal{N}(v)$ returns the neighbors of $u$, $h_\Theta$ is a neural network, and $\varepsilon$ is a constant. We implement $h_\Theta$ as a 2-layer Multi-Layer Perceptron ([10], MLP) with ReLU activation and set $\varepsilon = 0$. We obtain the final representations of nodes as $h_v = [h_v^{(1)}, \ldots, h_v^{(t)}]$ so that it contains the contextual information from 1-hop to $t$-hop. The graph-level representation is given by summation $h_\mathcal{G} = \sum_{v \in \mathcal{V}} h_v$. We use $h_\mathcal{G}$ to obtain the mean $\mu_\mathcal{G}$ and log variance $\sigma_\mathcal{G}$ of variational posterior approximation $q(z|\mathcal{G})$ through two separate linear layers and use the reparameterization trick [21] to sample from the distribution in the training process.

Given a latent variable $z$, our model first uses an autoregressive sequence generation model $P(\mathcal{F}_i|\mathcal{F}_{<i}, z)$ to decode an sequence of graph fragments $[\mathcal{F}_1, \ldots, \mathcal{F}_n]$. Note that the set of subgraphs into which a molecule is decomposed should not be ordered. Here, although we violate this law by using the ordered decoder, this issue is practically relieved by data augmentation of random fragment permutations during training, and it is further mitigated by the second step (presented later) where the assembling of the predicted subgraphs is not order-sensitive. During training, we insert two special tokens "`<start>`" and "`<end>`" indicating the beginning and the end of sequence generation, respectively. We implement the sequence model via a single layer of GRU [8] and project the latent variable $z$ to the initial state of GRU. The training objective of this stage is to minimize the log-likelihood $\mathcal{L}_\mathcal{P}$ of the ground truth fragment sequence:

$$\mathcal{L}_\mathcal{F} = \sum_{i=1}^{n} -\log P(\mathcal{F}_i|\mathcal{F}_{<i}, z). \qquad (3)$$

During testing, the generation stops when a "`<end>`" is outputted.

**How to assemble.**   This step estimates all inter-fragment edges $\{\mathcal{E}_{ij}\}$ non-autoregressively and globally. Specifically, we use a GNN with the same architecture in the last step but with different parameters to obtain the representations $h_v$ of each atom $v$ in all fragments. Given nodes $v$ and $u$ in two different fragments, we predict their connections as follows:

$$P(e_{uv}|z) = H_\theta([h_v; h_u; z]), \qquad (4)$$

where $H_\theta$ is a 3-layer MLP with ReLU activation. Apart from the types of chemical bonds, we also add a special type "`<none>`" to the edge vocabulary which indicates there is no connection between two nodes. During training, we predict both $P(e_{uv})$ and $P(e_{vu})$ to let $H_\theta$ learn the undirected nature of chemical bonds. We use negative sampling [12] to balance the ratio of "`<none>`" and chemical bonds. Since only about 2% pairs of nodes has inter-fragment connections, negative sampling significantly improves the computational efficiency and scalability. The training objective of this stage is to minimize the log-likelihood:

$$\mathcal{L}_\mathcal{E} = \sum_{u \in \mathcal{F}_i, v \in \mathcal{F}_j, i \neq j} -\log P(e_{uv}|z). \qquad (5)$$

Joining the reconstruction losses of both steps (Eq. 3 and Eq. 5), we arrive at: $\mathcal{L}_{\text{rec}} = \mathcal{L}_\mathcal{F} + \mathcal{L}_\mathcal{E}$. When considering target property, we jointly train a 2-layer MLP from $z$ to predict the scores of target properties using the MSE loss. By denoting this extra loss as $\mathcal{L}_{\text{prop}}$, the final objective of our PS-VAE becomes:

$$\mathcal{L} = \alpha\mathcal{L}_{\text{rec}} + (1 - \alpha)\mathcal{L}_{\text{prop}} + \beta D_{\text{KL}}, \qquad (6)$$

where the the KL divergence $D_{\text{KL}}$ aligns the distribution of $z$ with the prior distribution $\mathcal{N}(0, I)$; $\alpha$ and $\beta$ balance the trade-off between different losses.

To decode $\{\mathcal{E}_{ij}\}$ in the inference phase, the decoder first assigns all possible inter-fragment connections with a bond type and a corresponding confidence level. We try to add bonds that have a

confidence level higher than $\delta_{\text{th}} = 0.5$ to the molecule in order of confidence level from high to low. For each attempt, we perform a valency and a cycle check to reject the connections that will cause violation of valency or form unstable rings which are too small or too large. Since this procedure may form unconnected graphs, we find the maximal connected component as the final result. The pseudo code for inference is deferred to Appendix D.[2]

## 4   Experiments

**Evaluation Tasks**   We first report the empirical results for the distribution-learning tasks in *GuacaMol benchmarks* [7] to evaluate if the model can generate realistic and diverse molecules. Then we validate our model on two common problems and the goal-directed tasks of the *GuacaMol benchmarks*. *Property Optimization* requires generating molecules with optimized properties. *Constrained Property Optimization* concentrates on improving the properties of given molecules with a restricted degree of modification. *GuacaMol Goal-Directed Benchmarks* consist of 20 molecular design tasks carefully curated by domain-experts [48, 43, 11].

**Baselines**   We compare our principal subgraph variational auto-encoder (**PS-VAE**) with the following state-of-the-art models. **JT-VAE** [18] is a variational auto-encoder that represents molecules as junction trees. It performs Bayesian optimization on the latent variable for property optimization. **GCPN** [47] combines reinforcement learning and graph representation for goal-directed molecular graph generation. **MRNN** [34] adopts two RNNs to autoregressively generate atoms and bonds respectively. It combines policy gradient optimization to generate molecules with desired properties. **GraphAF** [39] is a flow-based autoregressive model which is first pretrained for likelihood modeling and then fine-tuned with reinforcement learning for property optimization. **GraphDF** [28] is another flow-based autoregressive model but leverages discrete latent variables to better capture the discrete distribution of graphs. **GA** [32] adopts genetic algorithms for property optimization and models the selection of the subsequent population with a neural network. **HierVAE** [19] extracts fragments by breaking bridge bonds in molecules and uses them as building blocks. **FREED** [46] is an RL-based model and uses fragments from existing chemical libraries to construct the subgraph vocabulary. **MARS** [44] adopts Markov chain Monte Carlo sampling for generation and is the state-of-the-art approach on the GuacaMol goal-directed benchmarks. Since our generation model is universally compatible with molecular decomposition of non-overlapping fragments, we also integrate vocabularies from **HierVAE** and **FREED** into our two-step generation model, which are abbreviated as **H-VAE**[*] and **F-VAE**[*], respectively.

We use the ZINC250K [16] dataset for training, which contains 250,000 drug-like molecules up to 38 atoms. For GuacaMol benchmark, we add extra results on the QM9 [6, 37] dataset, which has 133,014 molecules up to 23 atoms. We choose $N = 300$ for property optimization and $N = 500$ for constrained property optimization. PS-VAE is trained for 6 epochs with a batch size of 32 and a learning rate of 0.001. We set $\alpha = 0.1$ and initialize $\beta = 0$. We adopt a warm-up method that increases $\beta$ by 0.002 every 1000 steps to a maximum of 0.01. More details are in Appendix G.

### 4.1   Results

**GuacaMol Distribution-Learning Benchmarks**   These benchmarks evaluate four metrics on 10,000 molecules generated by the models. Since all methods introduce validity check into generation, they can always generate chemically valid molecules. *Uniqueness* measures the ratio of unique ones in generated molecules. *Novelty* assesses the ability of the models to generate molecules not contained in the training set. *KL Divergence* evaluates the closeness between the training set molecules and the generated molecules in terms of the distributions of various physicochemical properties. *Fréchet ChemNet Distance (FCD)* calculates the closeness of the two sets of molecules with respect to their hidden representations in the ChemNet [35]. Each metric is normalized to 0 to 1, and a higher value indicates better performance. Table 1 shows the results of distribution-learning benchmarks on QM9 and ZINC250K. Our model achieves competitive results in all four metrics, which indicates our model can generate realistic molecules and prevent overfitting the training set. In addition, with the same vocabulary, H-VAE[*] surpasses HierVAE consistently, confirming the advantage of our two-step generation model over the one used in HierVAE. Moreover, even sharing the same two-step

---

[2]Codes for our PS-VAE are availabel at `https://github.com/THUNLP-MT/PS-VAE`.

decoder, PS-VAE is still better than H-VAE* and F-VAE* consistently, which is probably because our constructed vocabulary discovers more meaningful and reusable patterns than H-VAE* and F-VAE*. We have displayed some molecules sampled from the prior distribution in Appendix L.

Table 1: Results of GuacaMol distribution-learning benchmarks on QM9 and ZINC250K. Uniq, KL Div and FCD refer to Uniqueness, KL Divergence and Fréchet ChemNet Distance, respectively.

| Model | QM9 | | | | ZINC250K | | | |
|---|---|---|---|---|---|---|---|---|
| | Uniq($\uparrow$) | Novelty($\uparrow$) | KL Div($\uparrow$) | FCD($\uparrow$) | Uniq($\uparrow$) | Novelty($\uparrow$) | KL Div($\uparrow$) | FCD($\uparrow$) |
| JT-VAE | 0.549 | 0.386 | 0.891 | 0.588 | 0.988 | 0.988 | **0.882** | 0.263 |
| GCPN | 0.533 | 0.320 | 0.552 | 0.174 | 0.982 | 0.982 | 0.456 | 0.003 |
| GraphAF | 0.500 | 0.453 | 0.761 | 0.326 | 0.288 | 0.287 | 0.508 | 0.023 |
| GraphDF | 0.672 | **0.672** | 0.601 | 0.137 | **0.998** | **0.998** | 0.459 | 0.001 |
| GA | 0.008 | 0.008 | 0.429 | 0.004 | 0.008 | 0.008 | 0.705 | 0.001 |
| MARS | 0.659 | 0.612 | 0.547 | 0.123 | 0.737 | 0.737 | 0.798 | 0.271 |
| HierVAE | 0.416 | 0.285 | 0.802 | 0.426 | 0.131 | 0.131 | 0.602 | 0.001 |
| H-VAE* | 0.619 | 0.487 | 0.869 | 0.588 | 0.991 | 0.991 | 0.611 | 0.037 |
| F-VAE* | 0.466 | 0.400 | 0.889 | 0.490 | 0.988 | 0.988 | 0.676 | 0.085 |
| PS-VAE (ours) | **0.673** | 0.523 | **0.921** | **0.659** | 0.997 | 0.997 | 0.850 | **0.318** |

**Property Optimization** This task focuses on generating molecules with optimized Penalized logP ([23], PlogP) and QED [4]. PlogP is logP penalized by synthesis accessibility and ring size which has an unbounded range. QED measures the drug-likeness of molecules with a range of $[0, 1]$. Both properties are calculated by empirical prediction models [42, 4], and are widely used in previous works [18, 47, 39]. We constrain all models to generate molecules under 60 atoms to prevent them from utilizing the flaw of PlogP by keeping extending the carbon chain [39]. We first train a predictor on the latent space of previously-trained VAE models to simulate the scoring functions, then perform gradient ascending to search for optimized molecules [18, 27]. Hyperparameters can be found in Appendix G. Following previous works [18, 47, 39], we generate 10,000 optimized molecules and report the top-3 scores found by each model. Results in Table 2 show that our model surpasses the baselines consistently. Note that with the same vocabulary, F-VAE achieves much better results than FREED, which indicates the superiority of our two-step generation strategy to the one in FREED.

Table 2: Comparison of the top-3 property scores.

| Method | Penalized logP | | | QED | | |
|---|---|---|---|---|---|---|
| | 1st | 2nd | 3rd | 1st | 2nd | 3rd |
| JT-VAE | 5.30 | 4.93 | 4.49 | 0.925 | 0.911 | 0.910 |
| GCPN | 7.98 | 7.85 | 7.80 | **0.948** | 0.947 | 0.946 |
| MRNN | 8.63 | 6.08 | 4.73 | 0.844 | 0.796 | 0.736 |
| GraphAF | 12.23 | 11.29 | 11.05 | **0.948** | **0.948** | 0.947 |
| GraphDF | 13.70 | 13.18 | 13.17 | **0.948** | **0.948** | **0.948** |
| GA | 12.25 | 12.22 | 12.20 | 0.946 | 0.944 | 0.932 |
| MARS | 7.24 | 6.44 | 6.43 | 0.944 | 0.943 | 0.942 |
| FREED | 6.74 | 6.65 | 6.42 | 0.920 | 0.919 | 0.908 |
| H-VAE* | 11.41 | 9.67 | 9.31 | 0.947 | 0.946 | 0.946 |
| F-VAE* | 13.50 | 12.62 | 12.40 | **0.948** | **0.948** | 0.947 |
| PS-VAE (ours) | **13.95** | **13.83** | **13.65** | **0.948** | **0.948** | **0.948** |

**Constrained Property Optimization** This task refines molecular properties under the constraint that the Tanimoto similarity with Morgan fingerprint [36] is above a threshold $\delta$. As suggested by previous works [18, 47, 39], we optimize 800 molecules with the lowest PlogP in the test set of ZINC250K. Similar to the property optimization task, we perform gradient ascending on the latent variable with a max step of 80. We collect all latent variables which have better-predicted scores than the previous iteration and decode each of them 5 times, namely up to 400 molecules. Following Shi et al. [39], we initialize the generation with sub-graphs sampled from the original molecules. Then we choose the highest scoring one from the molecules that meet the similarity constraint. Table 3 shows our model can generate molecules with a higher PlogP score under the similarity constraints. Since our model uses subgraphs as building blocks, the degree of modification tends to be greater than atom-level models, leading to a lower success rate. Even so, the success rates by our model are still on par with the atom-level counterparts and are the best among all subgraph-level methods.

**GuacaMol Goal-Directed Benchmarks** To further explore the efficacy of PS-VAE, we provide the results for GuacaMol goal-directed benchmarks [7] in Table 4. All scores are continuous values between 0 and 1, and the higher, the better. In order to obtain more discernible results between

Table 3: Mean (standard deviation) on improvement in constrained property optimization.

| Model | $\delta = 0.2$ | | $\delta = 0.4$ | | $\delta = 0.6$ | |
|---|---|---|---|---|---|---|
| | Improvement | Success | Improvement | Success | Improvement | Success |
| JT-VAE | 1.68±1.85 | 97.1% | 0.84±1.45 | 83.6% | 0.21±0.71 | 46.4% |
| GCPN | 4.12±1.19 | 100% | 2.49±1.30 | 100% | 0.79±0.63 | 100% |
| GraphAF | 4.99±1.38 | 100% | 3.74±1.25 | 100% | 1.95±0.99 | 98.4% |
| GraphDF | 5.62±1.65 | 100% | 4.13±1.41 | 100% | 1.72±1.15 | 93.0% |
| GA | 3.04±1.60 | 100% | 2.34±1.34 | 100% | 1.35±1.06 | 95.9% |
| MARS | 4.13±1.23 | 100% | 2.41±0.76 | 99.2% | 1.21±0.64 | 69.6% |
| H-VAE* | 3.42±1.65 | 89.9% | 2.68±1.44 | 74.0% | 1.90±1.06 | 51.5% |
| F-VAE* | 4.82±1.57 | 98.9% | 3.60±1.39 | 89.9% | 2.40±1.17 | 68.6% |
| PS-VAE (ours) | **6.42±1.86** | 99.9% | **4.19±1.30** | 98.9% | **2.52±1.12** | 90.3% |

different approaches, we conduct experiments on QM9 and ZINC250K which have much less high-scoring molecules than ChEMBL [30] to make the tasks more challenging. Otherwise the majority of the benchmarks are easy to be optimized to the upper bound [1]. We adopt the same optimizing strategy as in the property optimization tasks. Hyperparameters can be found in Appendix G. Table 4 shows that our model remarkably performs better than H-VAE* and F-VAE*, and thus exhibits the benefit of our subgraph extraction method. The superior of H-VAE* to HierVAE indicates the advantage of our two-step assembling over the sequential sampling strategy.

Table 4: Results for GuacaMol goal-directed benchmarks. H-VAE*, F-VAE* and PS-VAE refer to VAE models with vocabularies from HierVAE, FREED and our principal subgraphs, respectively.

| benchmark | JT-VAE | GA | H-VAE* | F-VAE* | HierVAE | MARS | PS-VAE |
|---|---|---|---|---|---|---|---|
| *ZINC250K* | | | | | | | |
| Celecoxib rediscovery | 0.245 | 0.073 | 0.252 | 0.310 | 0.223 | 0.137 | **0.451** |
| Troglitazone rediscovery | 0.171 | 0.080 | 0.205 | 0.215 | 0.181 | 0.163 | **0.254** |
| Thiothixene rediscovery | 0.225 | 0.091 | 0.240 | 0.236 | 0.191 | 0.072 | **0.286** |
| Aripiprazole similarity | 0.351 | 0.240 | 0.382 | 0.390 | 0.240 | 0.350 | **0.432** |
| Albuterol similarity | 0.426 | 0.437 | 0.404 | 0.484 | 0.430 | 0.567 | **0.611** |
| Mestranol similarity | 0.278 | 0.337 | 0.360 | 0.377 | 0.281 | 0.297 | **0.515** |
| C11H24 | 0.004 | 0.244 | 0.014 | 0.050 | 0.050 | 0.124 | **0.634** |
| C9H10N2O2PF2Cl | 0.200 | 0.659 | 0.523 | 0.567 | 0.023 | 0.435 | **0.782** |
| Median molecules 1 | 0.161 | 0.190 | 0.191 | 0.204 | 0.115 | 0.167 | **0.263** |
| Median molecules 2 | 0.166 | 0.111 | 0.178 | 0.198 | 0.126 | 0.166 | **0.199** |
| Osimertinib MPO | 0.693 | 0.585 | 0.758 | 0.768 | 0.538 | 0.745 | **0.800** |
| Fexofenadine MPO | 0.607 | 0.495 | 0.655 | 0.685 | 0.523 | 0.682 | **0.715** |
| Ranolazine MPO | 0.330 | 0.280 | 0.411 | 0.532 | 0.305 | 0.552 | **0.688** |
| Perindopril MPO | 0.370 | 0.239 | 0.394 | 0.403 | 0.361 | 0.418 | **0.429** |
| Amlodipine MPO | 0.442 | 0.413 | 0.477 | 0.463 | 0.341 | 0.477 | **0.544** |
| Sitagliptin MPO | 0.160 | 0.243 | 0.246 | 0.350 | 0.092 | 0.261 | **0.439** |
| Zaleplon MPO | 0.393 | 0.259 | 0.394 | 0.450 | 0.268 | 0.329 | **0.495** |
| Valsartan SMARTS | 0 | 0 | 0 | 0 | 0 | 0 | $5.398 \times 10^{-12}$ |
| Deco Hop | 0.576 | 0.530 | 0.577 | 0.577 | 0.559 | 0.572 | **0.601** |
| Scaffold Hop | 0.440 | 0.375 | 0.443 | 0.450 | 0.417 | 0.442 | **0.486** |
| *QM9* | | | | | | | |
| Celecoxib rediscovery | 0.056 | 0.106 | 0.111 | 0.129 | 0.059 | 0.023 | **0.210** |
| Troglitazone rediscovery | 0.070 | 0.110 | 0.124 | 0.131 | 0.121 | 0.085 | **0.152** |
| Thiothixene rediscovery | 0.047 | 0.061 | 0.115 | 0.100 | 0.059 | 0.037 | **0.146** |
| Aripiprazole similarity | 0.099 | 0.095 | 0.167 | 0.178 | 0.167 | 0.104 | **0.214** |
| Albuterol similarity | 0.301 | 0.243 | 0.305 | 0.277 | 0.380 | 0.305 | **0.463** |
| Mestranol similarity | 0.170 | 0.134 | 0.157 | 0.122 | **0.234** | 0.233 | 0.198 |
| C11H24 | $9.734 \times 10^{-7}$ | $2.148 \times 10^{-7}$ | $6.899 \times 10^{-7}$ | $2.979 \times 10^{-6}$ | 0.007 | $1.959 \times 10^{-45}$ | **0.015** |
| C9H10N2O2PF2Cl | 0.182 | 0.066 | 0.236 | 0.109 | 0.250 | 0.190 | **0.381** |
| Median molecules 1 | 0.226 | 0.185 | 0.195 | 0.194 | 0.229 | 0.143 | **0.261** |
| Median molecules 2 | 0.063 | 0.062 | 0.091 | 0.091 | 0.081 | 0.085 | **0.112** |
| Osimertinib MPO | 0.408 | 0.556 | 0.541 | 0.554 | 0.561 | 0.555 | **0.636** |
| Fexofenadine MPO | 0.217 | 0.403 | 0.307 | 0.290 | 0.355 | **0.443** | 0.373 |
| Ranolazine MPO | 0.004 | **0.170** | 0.007 | 0.010 | 0.011 | 0.032 | 0.018 |
| Perindopril MPO | 0.104 | 0.061 | 0.166 | 0.165 | 0.116 | 0.145 | **0.205** |
| Amlodipine MPO | 0.192 | 0.166 | 0.258 | 0.211 | 0.300 | 0.284 | **0.303** |
| Sitagliptin MPO | $2.556 \times 10^{-6}$ | $1.688 \times 10^{-6}$ | $2.807 \times 10^{-5}$ | $5.114 \times 10^{-5}$ | $0.951 \times 10^{-4}$ | $0.138 \times 10^{-2}$ | $0.319 \times 10^{-3}$ |
| Zaleplon MPO | $6.564 \times 10^{-5}$ | 0.03 | $3.41 \times 10^{-5}$ | $5.647 \times 10^{-5}$ | $0.771 \times 10^{-4}$ | 0.033 | **0.085** |
| Valsartan SMARTS | 0 | 0 | 0 | 0 | 0 | 0 | 0 |
| Deco Hop | 0.505 | 0.525 | 0.506 | 0.492 | 0.512 | 0.513 | **0.545** |
| Scaffold Hop | 0.341 | 0.361 | 0.340 | 0.354 | 0.351 | 0.362 | **0.393** |

## 4.2 Ablation Study

We conduct an ablation study to further validate the effects of principal subgraphs and the two-step generation approach. We first downgrade the vocabulary to contain only single atoms. Then we replace the two-step decoder with a fully autoregressive decoder. We present the performance on (constrained) property optimization in Table 5 and Table 6. The direct introduction of the two-step generation approach leads to improvement in property optimization but harms the performance on constrained property optimization. This is reasonable as separating the generation of atoms and bonds brings in information loss of bonds for the atom-level generation process. However, the

adoption of principal subgraphs as building blocks alleviates this negative effect since the principal subgraphs themselves contain rich bond information. Therefore, integrating principal subgraphs with our two-step generation approach leads to the best choice in our model design. Furthermore, we provide the runtime cost of baselines and PS-VAE in Appendix F to demonstrate the improvement in efficiency brought by the principal subgraphs and the two-step generation approach.

Table 5: Comparison of the top-3 property scores found by PS-VAE without certain modules

| Method | Penalized logP | | | QED | | |
|---|---|---|---|---|---|---|
| | 1st | 2nd | 3rd | 1st | 2nd | 3rd |
| PS-VAE | **13.95** | **13.83** | **13.65** | **0.948** | **0.948** | **0.948** |
| - PS | 6.91 | 5.50 | 5.12 | 0.870 | 0.869 | 0.869 |
| - two-step | 3.54 | 3.54 | 3.22 | 0.737 | 0.734 | 0.729 |

Table 6: Comparison of PS-VAE without certain modules on constrained property optimization.

| Model | $\delta = 0.2$ | | $\delta = 0.4$ | | $\delta = 0.6$ | |
|---|---|---|---|---|---|---|
| | Improvement | Success | Improvement | Success | Improvement | Success |
| PS-VAE | **6.42±1.86** | 99.9% | **4.19±1.30** | 98.9% | **2.52±1.12** | 90.3% |
| - PS | 2.33±1.46 | 74.8% | 2.12±1.36 | 50.9% | 1.87±1.12 | 27.0% |
| - two-step | 3.36±1.58 | 98.6% | 2.72±1.24 | 82.0% | 1.88±1.05 | 45.1% |

# 5 Analysis

**Principal Subgraph Statistics** We compare the distributions of the vocabulary of JT-VAE, FREED, HierVAE, and the vocabulary constructed by our principal subgraph extraction methods with a size of 100, 300, 500, and 700. Figure 4 shows the proportion of fragments with different numbers of atoms in the vocabulary and their frequencies of occurrence in the ZINC250K dataset. The fragments in the vocabulary of JT-VAE mainly contain 5 to 8 atoms with a sharp distribution. However, starting from fragments with 3 atoms, the frequency of occurrence is already close to zero. Therefore, the majority of fragments in the vocabulary of JT-VAE are actually not common fragments. On the contrary, the fragments in the vocabulary of principal subgraphs have a relatively smooth distribution over 4 to 10 atoms. Moreover, these fragments also have a much higher frequency of occurrence compared to those by JT-VAE. Vocabularies constructed by our method show similar advantages compared to those of HierVAE and FREED. We present samples of principal subgraphs in Appendix K.

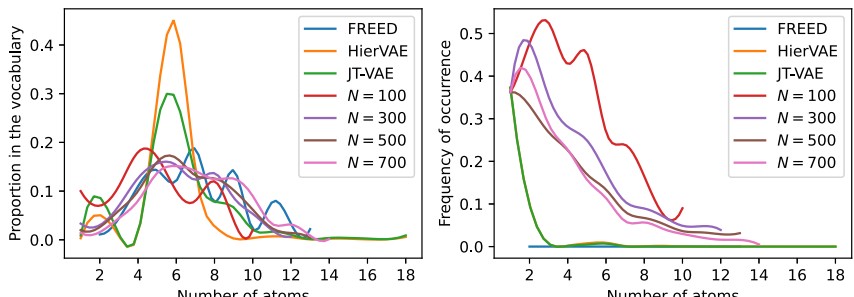

Figure 4: The left and right figures show the proportion of and frequency of occurrence of fragments with different number of atoms in the vocabulary, respectively.

**Principal Subgraph - Property Correlation** To analyze the principal subgraph - property correlation and whether our model can discover and utilize the correlation, we present the normalized distribution of generated fragments and Pearson correlation coefficient between the fragments and Penalized logP (PlogP) in Figure 5. The curve indicates that some fragments positively correlate with PlogP while some negatively correlate with it. Compared with the flat distribution under the non-optimization setting, the generated distribution shifts towards the fragments positively correlated with PlogP under the PlogP-optimization setting. The generation of fragments negatively correlated with PlogP is also suppressed. Therefore, correlations exist between fragments and PlogP, and our model can accurately discover and utilize these correlations.

**Proper Granularity** A larger $N$ in the principal subgraph extraction process leads to an increase in the number of atoms in extracted fragments and a decrease in their frequency of occurrence, as illustrated in Figure 6. These two factors affect model performance in opposite ways. On the one hand, the entropy of the dataset decreases with more coarse-grained decomposition [29], which benefits

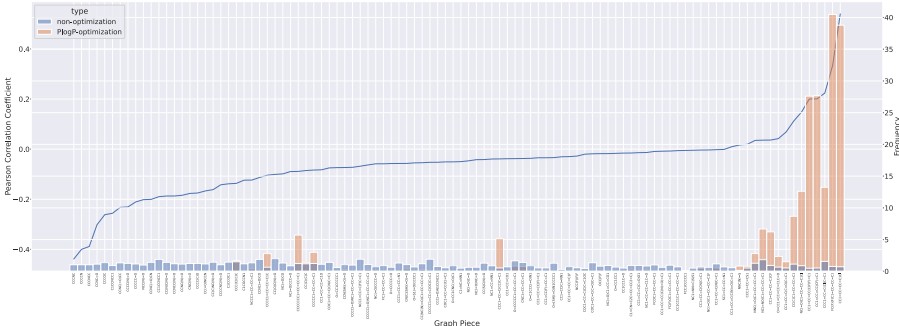

Figure 5: The distributions of generated fragments with and without optimization of PlogP, as well as Pearson correlation coefficient between the fragments and the score of PlogP. PlogP refers to Penalized logP. The distributions are normalized by the distribution of the training set, which means the frequency of occurrence of a fragment is divided by its count of occurrence in the training set.

model learning [3]. On the other hand, the sparsity problem worsens as the frequency of fragments decreases, which hurts model learning [2]. We propose a quantified method to balance entropy and sparsity. The entropy of the dataset given a set of fragments $\mathbb{V}$ is defined by the sum of the entropy of each fragment normalized by the average number of atoms: $H_{\mathbb{V}} = -\frac{1}{n_{\mathbb{V}}} \sum_{\mathcal{F} \in \mathbb{V}} P(\mathcal{F}) \log P(\mathcal{F})$, where $P(\mathcal{F})$ is the relative frequency of fragment $\mathcal{F}$ in the dataset and $n_{\mathbb{V}}$ is the average number of atoms of fragments in $\mathbb{V}$. The sparsity of $\mathbb{V}$ is defined as the reciprocal of the average frequency of fragments $f_{\mathbb{V}}$ normalized by the size of the dataset $M$: $S_{\mathbb{V}} = M/f_{\mathbb{V}}$. Then the entropy - sparsity trade-off ($T$) can be expressed as: $T_{\mathbb{V}} = H_{\mathbb{V}} + \gamma S_{\mathbb{V}}$, where $\gamma$ balances the impacts of entropy and sparsity since the impacts vary across different tasks. We assume that $T_{\mathbb{V}}$ negatively correlates with downstream tasks. Given a task, we first sample several values of $N$ to calculate their values of $T$ and then compute the $\gamma$ that minimize the Pearson correlation coefficient between $T$ and the corresponding performance on the task. With the proper $\gamma$, Pearson correlation coefficients for the first three downstream tasks in this paper are -0.987, -0.999, and -0.707, indicating strong negative correlations. For example, Figure 6 shows the curve of entropy - sparsity trade-off with a maximum of 3,000 iteration steps for the property optimization task. From the curve, we choose $N = 300$ for the property optimization task. Please refer to Appendix B for more experimental details.

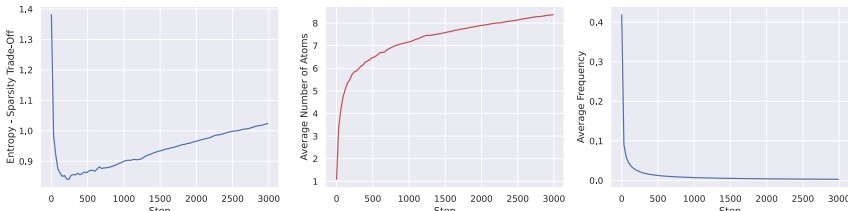

Figure 6: Entropy - Sparsity trade-off, average number of atoms in principal subgraphs and average frequency of occurrence of principal subgraphs with a maximum of 3,000 iteration steps.

## 6   Conclusion

We propose an algorithm to automatically discover the regularity in molecules and extract them as *principal subgraphs*. With the extracted principle subgraphs at hand, we generate molecules in two phases. Our model consistently outperforms state-of-the-art models on distribution-learning, (constrained) property optimization, and GuacaMol goal-directed benchmarks, and exhibits higher computational efficiency than certain widely-used baselines. Our work provides insights into the selection of subgraphs on molecular graph generation and can inspire future search in this direction.

## Acknowledgments and Disclosure of Funding

This work is jointly supported by the National Natural Science Foundation of China (No.61925601, No.62006137); Guoqiang Research Institute General Project, Tsinghua University (No. 2021GQG1012); Beijing Academy of Artificial Intelligence; Huawei Noah's Ark Lab; Beijing Outstanding Young Scientist Program (No. BJJWZYJH012019100020098).

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
