# A Proof of Theorem 3.2

**Theorem A.1.** *The vocabulary $\mathbb{V}$ constructed by Algorithm 1 exhibits the following advantageous properties.*

> *(i) **Monotonicity**: The frequency of the non-single-atom fragments in $\mathbb{V}$ decreases monotonically, namely, $\forall \mathcal{F}_i, \mathcal{F}_j \in \mathbb{V}, c(\mathcal{F}_i) \leq c(\mathcal{F}_j)$, if $i \geq j$.*

> *(ii) **Significance**: Each fragment $\mathcal{F}$ in $\mathbb{V}$ is a principal subgraph.*

> *(iii) **Completeness**: For any principal subgraph $\mathcal{S}$ arising in the dataset, there always exists a fragment $\mathcal{F}$ in $\mathbb{V}$ satisfying $\mathcal{S} \subseteq \mathcal{F}, c(\mathcal{S}) = c(\mathcal{F})$, when $\mathbb{V}$ has collected all fragments with frequency no less than $c(\mathcal{S})$.*

*Proof.* Prior to the proof, we first present a clear observation of the created vocabulary $\mathbb{V}$:

**Proposition A.2.** *Given any $\mathcal{F}, \mathcal{F}' \in \mathbb{V}$, for any their instances arising on an arbitrary molecule during the extraction process, either they are not spatially intersected $\mathcal{F} \cap \mathcal{F}' = \emptyset$, or they contain each other: $\mathcal{F} \subseteq \mathcal{F}'$ or $\mathcal{F}' \subseteq \mathcal{F}$.*

Now we prove each claim in the above theorem.

**(i) Monotonicity**. We prove it by contradiction. Suppose that there exist $\mathcal{F}_{i_1}, \mathcal{F}_{i_2} \in \mathbb{V}, i_1 > i_2$, and $c(\mathcal{F}_{i_1}) > c(\mathcal{F}_{i_2})$. According to Proposition A.2, we have either $\mathcal{F}_{i_1} \subseteq \mathcal{F}_{i_2}$ or $\mathcal{F}_{i_1} \cap \mathcal{F}_{i_2} = \emptyset$ on each molecular. If it is the former case, then $\mathcal{F}_{i_1}$ should be firstly extracted and then merged with other fragments to yield $\mathcal{F}_{i_2}$ which means $i_1 < i_2$, conflicting with the assumption. If it is the latter case, since $c(\mathcal{F}_{i_1}) > c(\mathcal{F}_{i_2})$ implying that $\mathcal{F}_{i_1}$ is first extracted, also conflicting with the assumption we set. Hence, the claim is proved.

**(ii) Significance**. It is proved by contradiction as well. For any fragment $\mathcal{F} \in \mathbb{V}$ on a certain molecule, suppose we have a subgraph $\mathcal{S}$ satisfying $\mathcal{S} \cap \mathcal{F} \neq \emptyset, c(\mathcal{S}) > c(\mathcal{F})$, and $\mathcal{S} \nsubseteq \mathcal{F}$. Then we must find two connected nodes $v_1$ and $v_2$, where $v_1$ belongs to $\mathcal{S}$ but not $\mathcal{F}$: $v_1 \in \mathcal{S}, v_1 \notin \mathcal{F}$, and $v_2$ belongs to both subgraphs: $v_2 \in \mathcal{S} \cap \mathcal{F}$. Let us construct a new subgraph with two nodes $\mathcal{S}' = \{v_1, v_2, e_{12}\}$ where $e_{12}$ is the edge connecting $v_1$ and $v_2$. Obviously, $c(\mathcal{S}') \geq c(\mathcal{S}) > c(\mathcal{F})$, implying that $\mathcal{S}'$ has been included into the vocabulary $\mathbb{V}$ before $\mathcal{F}$ is generated. However, by its definition, $\mathcal{S}' \cap \mathcal{F} \neq \emptyset$, $\mathcal{S}' \nsubseteq \mathcal{F}$, and $\mathcal{F} \nsubseteq \mathcal{S}$ which makes a contradiction with Proposition A.2. Thus, the assumption fails, and the claim is proved.

**(iii) Completeness**. Without loss of generality, we suppose $\mathcal{S}$ contains at least two nodes. We choose an arbitrary node from $\mathcal{S}$, then we expand it during the vocabulary conduction via Algorithm 1. We keep merging this node with the fragments in $\mathbb{V}$ to produce $\mathcal{F}_t$ at each iteration $t$ until the following cases happen: 1) $\mathcal{F}_t = \mathcal{S}$, which directly leads to the conclusion of the claim; 2) it is the first time we merge the current fragment $\mathcal{F}_{t-1} \subsetneq \mathcal{S}$ with an external fragment $\mathcal{F}' \nsubseteq \mathcal{S}$ to yield $\mathcal{F}_t$. Now, we solely discuss the second case. On one hand, if $c(\mathcal{F}_t) > c(\mathcal{S})$, we can always find a subgraph consisting of two connected nodes from $\mathcal{F}'$ and $\mathcal{F}_{t-1}$, respectively, and this subgraph contains at least one node not in $\mathcal{S}$ and has a larger frequency than $\mathcal{S}$, which conflicts with the condition that $\mathcal{S}$ is a principal subgraph. On the other hand, if $c(\mathcal{F}_t) < c(\mathcal{S})$, we can always find a node in $\mathcal{S}$ but not in $\mathcal{F}_{t-1}$ merged with with $\mathcal{F}_{t-1}$ to generate a fragment with a larger frequency than $\mathcal{F}'$, which also conflicts with the implementation rule of Algorithm 1 since $\mathcal{F}'$ is of the largest frequency among all potential merging choices with $\mathcal{F}_{t-1}$. Hence, we only have $c(\mathcal{F}_t) = c(\mathcal{S})$, based on which, we can always merge $\mathcal{F}_t$ with the remaining part of $\mathcal{S}$ at later iterations to make $\mathcal{F}_t \supseteq \mathcal{S}$ while keeping $c(\mathcal{F}_t) = c(\mathcal{S})$. The proof is concluded. $\qquad\square$

# B Effects of Different Sizes of Vocabulary

In Table 7 and Table 8, we also provide the results for (constrained) property optimizationof PS-VAE with $N = 100, 300, 500, 700$ for more direct illustration.

We observe that the best values of $N$ for the above two tasks are 300 and 500, respectively, both of which are consistent with the optimal points by the trade-off curves in Figure 6 of Section 5. It suggests that in practice we can tune the value of $N$ by the method proposed in Section 5.

Table 7: Comparison of the top-3 property scores found by PS-VAE with different vocabulary size.

| N | Penalized logP | | | QED | | |
|---|---|---|---|---|---|---|
| | 1st | 2nd | 3rd | 1st | 2nd | 3rd |
| 100 | 10.30 | 10.06 | 9.96 | 0.9480 | 0.9478 | 0.9478 |
| 300 | 13.95 | 13.83 | 13.65 | **0.9483** | **0.9482** | **0.9482** |
| 500 | 12.57 | 12.24 | 12.21 | **0.9483** | **0.9482** | 0.9480 |
| 700 | 8.41 | 8.20 | 8.10 | 0.9482 | 0.9481 | 0.9480 |

Table 8: Mean (standard deviation) on improvement in constrained property optimization of PS-VAE with different vocabulary size.

| N | $\delta = 0.2$ | | $\delta = 0.4$ | | $\delta = 0.6$ | |
|---|---|---|---|---|---|---|
| | Improvement | Success | Improvement | Success | Improvement | Success |
| 100 | 5.17±1.66 | 99.5% | 3.65±1.30 | 95.9% | 2.30±1.04 | 79.1% |
| 300 | 5.42±2.30 | 99.4% | 3.70±1.54 | 94.4% | 2.31±1.12 | 75.2% |
| 500 | **6.42±1.86** | 99.9% | **4.19±1.30** | 98.9% | **2.52±1.12** | 90.3% |
| 700 | 4.93±1.81 | 99.6% | 3.61±1.37 | 93.8% | 2.26±1.13 | 72.8% |

## C  Subgraph-Level Decomposition Algorithm

Algorithm 2 presents the pseudo code for the subgraph-level decomposition of molecules. The algorithm takes the atom-level molecular graph, the vocabulary of principal subgraphs, and their frequencies of occurrence recorded during the principal subgraph extraction process as input. Then the algorithm iteratively merges the two neighboring principal subgraphs whose union has the highest recorded frequency of occurrence in the vocabulary until all possible unions of two neighboring principal subgraphs are not in the vocabulary. We provide further illustrations for the mechanism of "MergeSubGraph" as follows. It takes as input each molecular $\mathcal{G}$ and the selected top-1 fragment $\mathcal{F}$. If $\mathcal{G}$ contains $\mathcal{F}$, then we will merge the two adjacent nodes in $\mathcal{G}$ that comprise $\mathcal{F}$ into a new node $\mathcal{F}$.

---

**Algorithm 2** Subgraph-Level Decomposition

---

**Input:** A graph $\mathcal{G}$ that decomposed into atoms, the set $\mathbb{V}$ of learned principal subgraphs, and the counter $\mathcal{C}$ of learned principal subgraphs.
**Output:** A new representation $\mathcal{G}'$ of $\mathcal{G}$ that consists of principal subgraphs in $\mathbb{V}$.
$\mathcal{G}' \leftarrow \mathcal{G}$;
**while** True **do**
   $freq \leftarrow -1; \mathcal{F} \leftarrow None$;
   **for** $\langle \mathcal{F}_i, \mathcal{F}_j, \mathcal{E}_{ij} \rangle$ **in** $\mathcal{G}'$ **do**
      $\mathcal{F}' \leftarrow \text{Merge}(\langle \mathcal{F}_i, \mathcal{F}_j, \mathcal{E}_{ij} \rangle)$; {*Merge neighboring fragments into a new fragment.*}
      $s \leftarrow \text{GraphToSMILES}(\mathcal{F}')$; {*Convert a graph to SMILES representation.*}
      **if** $s$ in $\mathbb{V}$ and $\mathcal{C}[s] > freq$ **then**
         $freq \leftarrow \mathcal{C}[s]$;
         $\mathcal{F} \leftarrow \mathcal{F}'$;
      **end if**
   **end for**
   **if** $freq == -1$ **then**
      break;
   **else**
      $\mathcal{G}' \leftarrow \text{MergeSubGraph}(\mathcal{G}', \mathcal{F})$; {*Update the graph representation.*}
   **end if**
**end while**

---

## D    Inference Algorithm for Bond Completion

Algorithm 3 shows the pseudo code of our inference algorithm. We first predict the bonds between all possible pairs of atoms in which the two atoms are in different fragments and sort them by the confidence level given by the model from high to low. Then for each bond with a confidence level higher than the predefined threshold $\delta_{th}$, which is 0.5 in our experiments, we add it into the molecular graph if it passes the valence check and cycle check. The valence check ensures the given bond will not violate valence rules. The cycle check ensures the given bond will not form unstable rings with nodes less than 5 or more than 6.

---

**Algorithm 3** Inference Algorithm for Bond Completion

---

**Input:** An incomplete molecular graph $\mathcal{G}$ composed of fragments where inter-fragment bonds are absent, the predicted bond type for all possible inter-fragment connections $\mathcal{B}$ and the map to their confidence level $\mathcal{C}$, the threshold for confidence level $\delta_{th}$
**Output:** A valid molecular graph $\mathcal{G}'$
$\mathcal{G}' \leftarrow \mathcal{G}$;
$\mathcal{B} \leftarrow \mathrm{SortByConfidence}(\mathcal{B}, \mathcal{C})$; {*Sort the bonds by their confidence level from high to low.*}
**for** $b_{uv}$ **in** $\mathcal{B}$ **do**
    **if** $\mathcal{C}[b_{uv}] < \delta_{th}$ **then**
        continue; {*Discard edges with confidence level lower than the threshold.*}
    **end if**
    **if** valence_check($b_{uv}$) **and** cycle_check($b_{uv}$) **then**
        $\mathcal{G}' \leftarrow \mathrm{AddEdge}(\mathcal{G}', b_{uv})$;{*Add edges that pass valence and cycle check to $\mathcal{G}'$*}
    **end if**
**end for**
$\mathcal{G}' \leftarrow \mathrm{MaxConnectedComponent}(\mathcal{G}')$; {*Find the maximal connected component in $\mathcal{G}'$*}

---

## E    Complexity Analysis

**Principal Subgraph Extraction**    The GraphToSMILES function is implemented with CAN-GEN [41] algorithm, which tackles the conversion from molecular graph to SMILES with a complexity of $O(n^2 \log n)$, where $n$ denotes the number of atoms in a molecule. The SMILESToGraph function constructs a molecular graph from a given SMILES with $O(n)$ complexity. Particularly for small molecules, the largest number of atoms $n$ is restricted by the molecular weight, hence we can practically assume these two functions have constant running time. Since the number of two neighboring fragments equals the number of inter-fragment connections in the subgraph-level graph, the complexity is $O(NMe)$, where $N$ is the predefined size of vocabulary, $M$ denotes the number of molecules in the dataset, and $e$ denotes the maximal number of inter-fragment connections in a single molecule. The number of inter-fragment connections decreases rapidly in the first few iterations, therefore the time cost for each iteration decreases rapidly. It cost 6 hours to perform 500 iterations on 250,000 molecules in the ZINC250K dataset with 4 CPU cores.

**Subgraph-Level Decomposition**    Given an arbitrary molecule, the worst case is that each iteration adds one atom to one existing fragment until the molecule is finally merged into a single fragment. In this case, the algorithm runs for $|\mathbb{V}|$ iterations. Therefore, the complexity is $O(|\mathbb{V}|)$ where $\mathbb{V}$ includes all the atoms in the molecule.

## F    Runtime Cost

We train JT-VAE, GraphAF, and our PS-VAE on a machine with 1 NVIDIA GeForce RTX 2080Ti GPU and 32 CPU cores to compare their efficiency of training and inference. All models are trained over a fixed number of epochs (*i.e.* 6) and then generate 10,000 molecules. As shown in Table 9, our model achieves significant improvements in efficiency due to subgraph-based two-step generation. With principal subgraphs as building blocks, the number of steps required to generate a molecule is significantly decreased compared to the atom-level models like GraphAF. Moreover, since the two-step generation approach separates the generation of principal subgraphs and the assembling of them into two stages, it formalizes the bond completion as a link prediction task and avoids the

exhausting enumeration of all possible combinations adopted by JT-VAE. Therefore, our model achieves tremendous improvement in computational efficiency over these baselines.

Table 9: Runtime cost for JT-VAE, GraphAF and our PS-VAE on the ZINC250K dataset. Inference time is measured with the generation of 10,000 molecules. Avg Step denotes the average number of steps each model requires to generate a molecule.

| Model | Training | Inference | Avg Step |
|---|---|---|---|
| JT-VAE | 24 hours | 20 hours | 15.50 |
| GCPN | 14 hours | 20 hours | 38.21 |
| GraphAF | 7 hours | 10 hours | 56.88 |
| HierVAE | 10.9 hours | 1.2 hours | 36.94 |
| PS-VAE (ours) | **1.2 hours** | **0.3 hour** | **6.84** |

# G   Experiment Details

**Model and Training Hyperparameters**   We present the choice of model parameters in Table 10 and training parameters in Table 11. We represent an atom with three features: atom embedding, fragment embedding and position embedding. Atom embedding is a trainable vector of size $e_{atom}$ for each type of atoms. Similarly, fragment embedding is a trainable vector of size $e_{fragment}$ for each type of fragment. Positions indicate the order of generation of fragments. For property optimization tasks, we jointly train a 2-layer MLP from the latent variable to predict property scores. For GuacaMol goal-directed benchmarks, the predictor is trained after the training of the VAE models. The training loss is represented as $\mathcal{L} = \alpha \cdot \mathcal{L}_{rec} + (1 - \alpha) \cdot \mathcal{L}_{prop} + \beta \cdot D_{KL}$ where $\alpha$ balances the reconstruction loss and prediction loss. For $\beta$, we adopt a warm-up method that increase it by $\beta_{stage}$ every fixed number of steps to a maximum of $\beta_{max}$. We found a $\beta$ higher than $0.01$ often causes KL vanishing problem and greatly harm the performance. Our model and the baselines are trained on the ZINC250K dataset with the same train / valid / test split as in Kusner et al. [23].

Table 10: Parameters in the principal subgraph variational auto-encoder

| Model | Param | Description | Value |
|---|---|---|---|
| Common | $e_{atom}$ | Dimension of embeddings of atoms. | 50 |
| | $e_{fragment}$ | Dimension of embeddings of fragments. | 100 |
| | $e_{pos}$ | Dimension of embeddings of postions. The max position is set to be 50. | 50 |
| Encoder | $d_h$ | Dimension of the node representations $\boldsymbol{h}_v$ | 300 |
| | $d_{\mathcal{G}}$ | The final representaion of graphs are projected to $d_{\mathcal{G}}$. | 400 |
| | $d_{\boldsymbol{z}}$ | Dimension of the latent variable. | 56 |
| | t | Number of iterations of GIN. | 4 |
| Decoder | $d_{GRU}$ | Hidden size of GRU. | 200 |
| Predictor | $d_p$ | Dimension of the hidden layer of MLP. | 200 |

Table 11: Training hyperparameters

| Param | Description | Value |
|---|---|---|
| $lr$ | Learning rate | 0.001 |
| $\alpha$ | Weight for balancing reconstruction loss and predictor loss | 0.1 |
| $\beta_{init}$ | Initial weight of KL Divergence | 0 |
| $\beta_{max}$ | Max weight of KL Divergence | 0.01 |
| $kl_{warmup}$ | The number of steps for one stage up in $\beta$ | 1000 |
| $\beta_{stage}$ | Increase of $\beta$ every stage | 0.002 |

**Property Optimization**   We use gradient ascending to search in the continuous space of latent variable. For simplicity, we set a target score and optimize the mean square error between the score given by the predictor and the target score just as in the training process. The optimization stops

if the mean square error does not drop for 3 iterations or it has been iterated to the $maxstep$. We normalize the Penalized logP in the training set to $[0, 1]$ according to the statistics of ZINC250K. By setting a target value higher than 1 the model is supposed to find molecules with better property than the molecules in the training set. To acquire the best performance, we perform a grid search with $lr \in \{0.001, 0.01, 0.1, 1, 2\}$, $maxstep \in \{20, 40, 60, 80, 100\}$ and $target \in \{1, 2, 3, 4\}$. For optimization of QED, we choose $lr = 0.01, maxstep = 100, target = 2$. For optimization of Penalized logP, we choose $lr = 0.1, maxstep = 100, target = 2$.

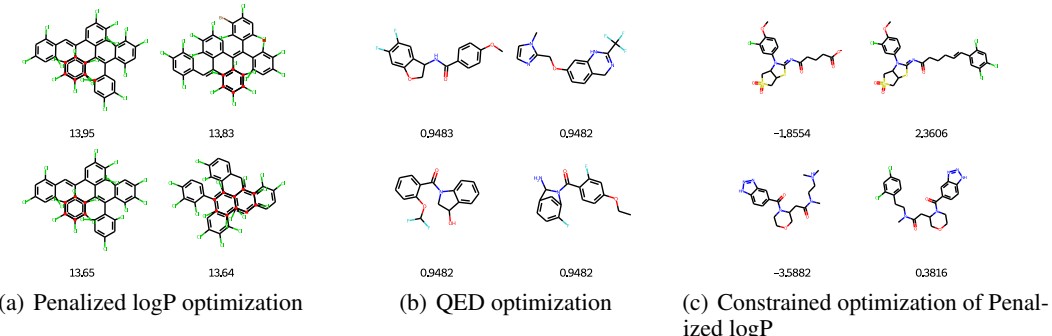

| (a) Penalized logP optimization | (b) QED optimization | (c) Constrained optimization of Penalized logP |

Figure 7: Samples of property optimization and constrained property optimization. In (c) the first and the second columns are the original and modified molecules labeled with their Penalized logP.

**Constrained Property Optimization**    We use the same method as property optimization to optimize the latent variable. We also perform a grid search with $lr \in \{0.1, 0.01\}$ and $target \in \{2, 3\}$. We select $lr = 0.1, maxstep = 80$ and $target = 2$. For decoding, we first initialize the generation with a submol sampled from the original molecule by teacher forcing. We follow Shi et al. [39] to first sample a BFS order of all atoms and then randomly drop out the last $m$ atoms with $m$ up to 5. We collect all latent variables which have better predicted scores than the previous iteration and decode each of them 5 times, namely up to 400 molecules. Then we choose the one with the highest property score from the molecules that meet the similarity constraint. For the baseline GA [1], we adjust the number of iterations to 5 and the size of population to 80, namely traversing up to 400 molecules, for fair comparison.

**GuacaMol Goal-directed Benchmarks**    After pretraining of our PS-VAE, we train an 2-layer MLP on the latent variable to predict all the properties in the benchmark. Then we do gradient ascending to search high-scoring molecules in the latent space as in the task of property optimization. We set $lr = 0.01, maxstep = 100, target = 2$. For iterative baselines (i.e. GA, MARS), we restrict the number of candidates they can explore to the same number as our PS-VAE does so that all methods are compared under the same searching efficiency.

## H    Fused Rings Generation

We conduct an additional experiment to validate the ability of PS-VAE to generate molecules with fused rings (cycles with shared edges), because at first thought it seems difficult for PS-VAE to handle these molecules due to the non-overlapping nature of principal subgraphs in a decomposed molecule. We train atom-level and subgraph-level PS-VAEs on all 4,431 structures consisting of fused rings from ZINC250K. Then we sample 1,000 molecules from the latent space to calculate the proportion of molecules with fused rings. The results are 94.5% and 97.2% for the atom-level model and the subgraph-level model, respectively. The experiment demonstrates that the introduction of principal subgraphs as building blocks will not hinder the generation of molecules with fused rings.

## I    Data Efficiency

Since the principal subgraphs are common subgraphs in the molecular graphs, they should be relatively stable with respect to the scale of training set. To validate this assumption, we choose

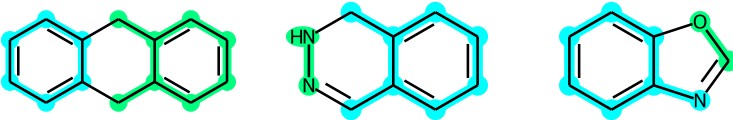

Figure 8: Decomposition of three molecules with fused rings (cycles that share edges).

subsets of different ratios to the training set for training to observe the trend of the coverage of Top 100 principal subgraphs in the vocabularies as well as the model performance on the average score of the distribution-learning benchmarks. As illustrated in Figure 9, with a subset above 20% of the training set, the constructed vocabulary covers more than 95% of the top 100 principal subgraphs in the full training set, as well as the model performance on the distribution-learning benchmarks.

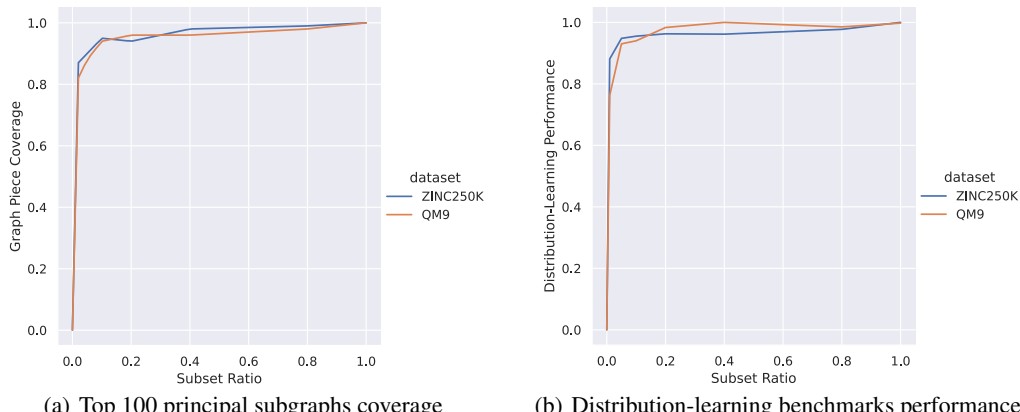

(a) Top 100 principal subgraphs coverage      (b) Distribution-learning benchmarks performance

Figure 9: The coverage of top 100 principal subgraphs and the relative performance on the distribution-learning benchmarks with respect to subsets of different ratios to the full training set.

## J  Discussion

**Universal Granularity Adaption**  The concept and extraction algorithm of *principal subgraphs* resemble those of subword units [38] in machine translation. Though subword units are designed for the out-of-vocabulary problem of machine translation, they also improve the translation quality [38]. In this work, we demonstrate the power of principal subgraphs and are curious about whether there is a universal way to adapt atom-level models into subgraph-level counterparts to improve their generation quality. The key challenge is to find an efficient and expressive way to encode inter-fragment connections into feature vectors. We leave this for future work.

**Searching in Continuous Space**  In recent years, reinforcement learning (RL) is becoming dominant in the field of optimization of molecular properties [47, 39]. These RL models usually suffer from reward sparsity when applied to multi-objective optimization [20]. However, most scenarios that incorporate molecular property optimization have multi-objective constraints (e.g., drug discovery). In this work, we show that with principal subgraphs, even a simple searching method like gradient ascending can surpass RL methods on single-objective optimization. It is possible that with better searching methods in continuous space our model can achieve competitive results on multi-objective optimization.

## K  Principal Subgraph Samples

We present 50 principal subgraphs found by our extraction algorithm in Figure 10.

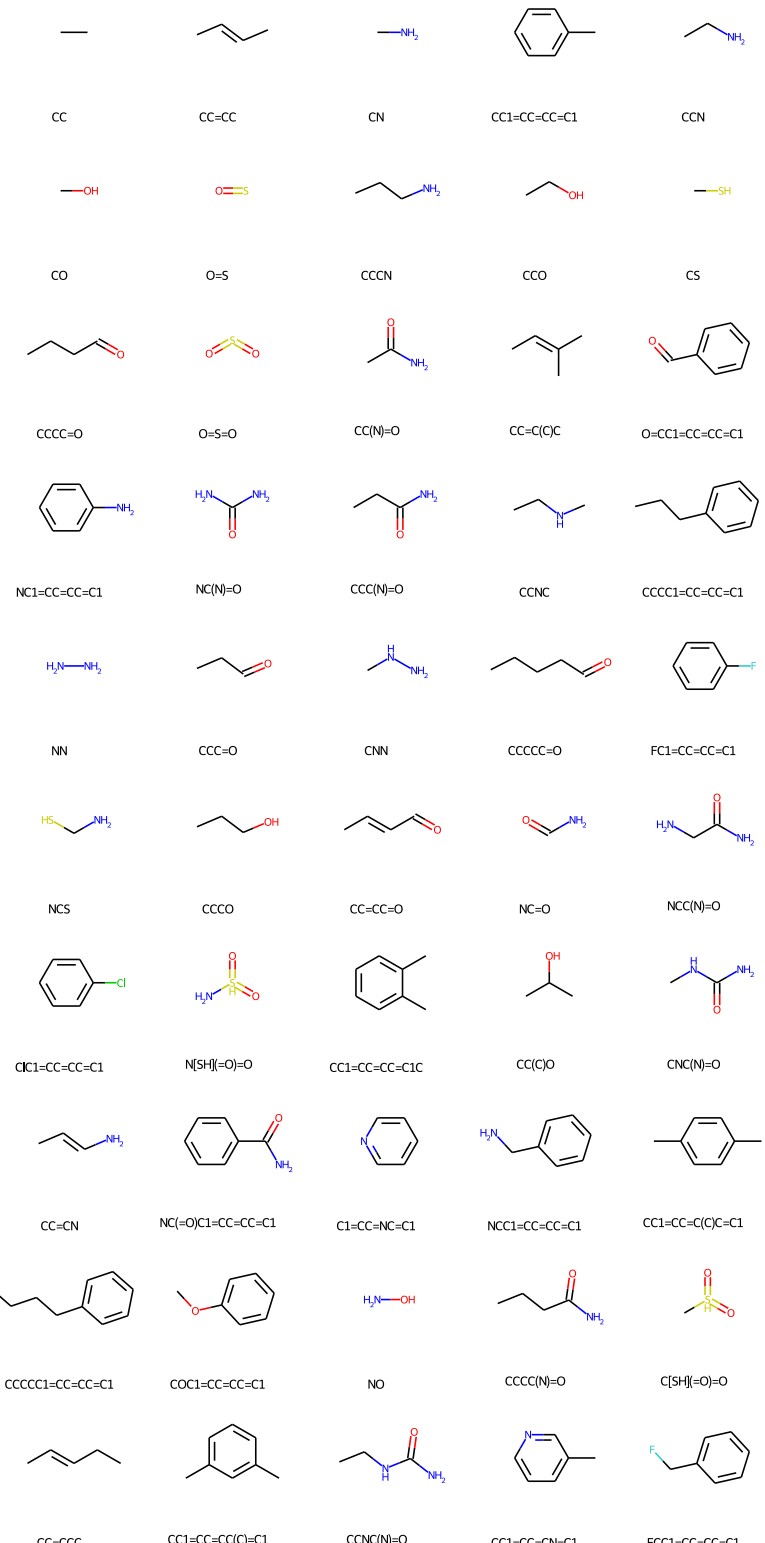

Figure 10: 50 Samples of principal subgraphs from the vocabulary with 100 principal subgraphs in total. Each principal subgraph is labeled with its SMILES representation.

## L More Molecule Samples

We further present 50 molecules sampled from the prior distribution in Figure 11.

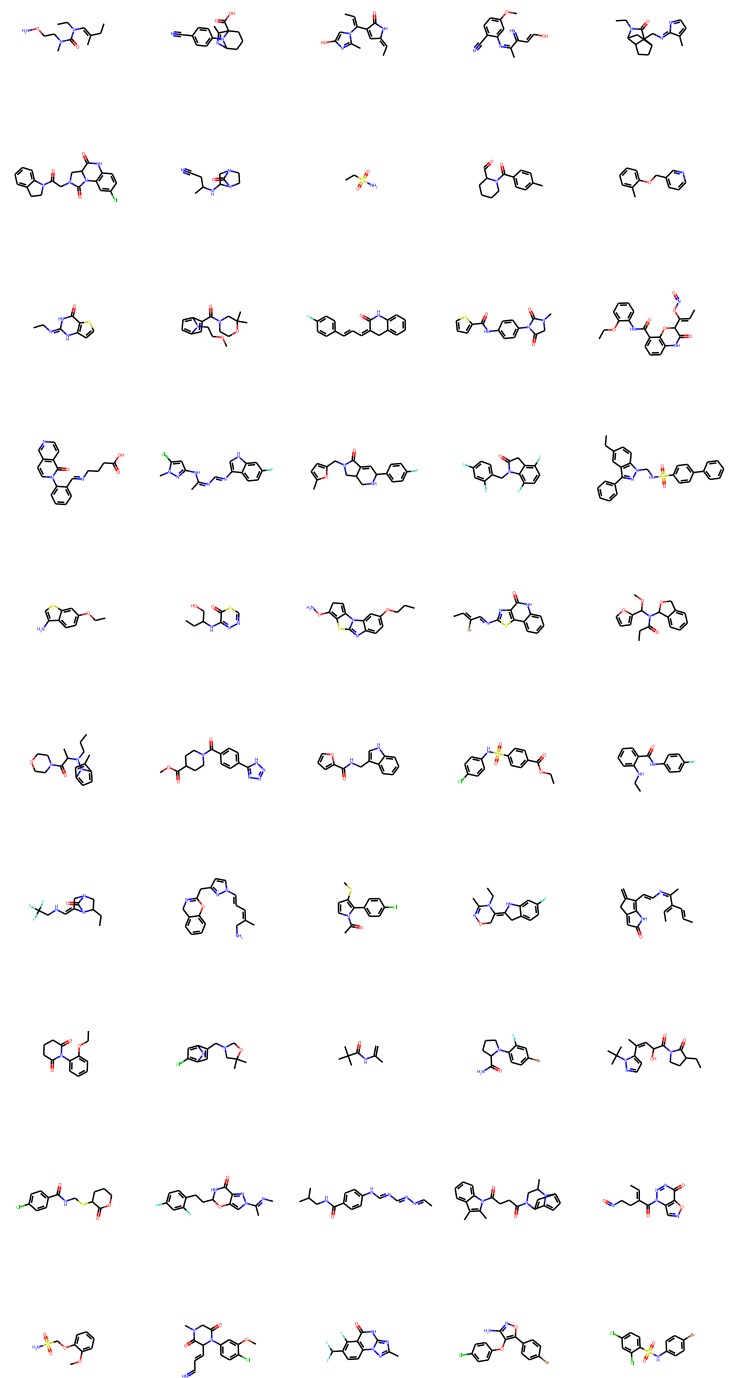

Figure 11: 50 molecules sampled from the prior distribution $\mathcal{N}(0, \boldsymbol{I})$