# OpenReview forum: "Molecule Generation by Principal Subgraph Mining and Assembling"
_NeurIPS.cc/2022/Conference — NeurIPS 2022 Accept_

### Official Review · Reviewer_EktL · 2022-07-08

**Rating:** 6
**Confidence:** 3
**Soundness:** 3 good
**Presentation:** 2 fair
**Contribution:** 3 good

**Summary:**

The authors propose a novel approach for molecule generation, introducing principal subgraphs. The method can be divided into two steps: The extraction of principal subgraphs from a training dataset and the molecule generation itself. In the first step, principal subgraphs are extracted from a training set by iteratively merging subgraphs, starting with single vertices. In each iteration, two neighboring subgraphs are merged, if the subgraph corresponding to their union is most frequent in the dataset among the possible merges.  For the molecule generation, first subgraphs and then edges between those subgraphs are predicted.

**Questions:**

* Q1) Are the generated molecules chemically valid and how is this guaranteed?
* Q2) Which implementation and parameter selection has been used for the different methods?
* Q3) Couly you provide additional results of the different reference methods?
    - FREED and MRNN in Tables 1, 3 and 4
    - MARS in Tables 1-3
    - JT-VAE, GCPN, GraphAF, and GA in Table 4

**Limitations:**

Overall these are addressed adequately. However, the limitations of principle subgraphs (e.g., compared to frequent subgraphs) could be explored in more detail.

**Strengths And Weaknesses:**

## Strengths
  * The paper proposes a new approach to molecule generation, the concept of principal subgraphs is nice. Related approaches are adequately cited.
  * The paper is well structured and written.

## Weaknesses
  * The concept of principal subgraphs is not explored in full detail. It would be nice to formalize the relation to frequent subgraphs. The authors state that principal subgraphs can be computed more efficiently than frequent subgraphs. Does this come with any costs/limitations, e.g., in terms of the completeness of the vocabulary? Moreover, a detailed complexity analysis would be very helpful to support this argument. The complexity analysis in the appendix is not sufficient. It appears to assume that functions like GraphToSMILES have constant running time, which needs justification. This function solves the graph canonization problem, for which no polynomial time algorithm is known in general.
  * The experimental evaluation is not fully convincing, e.g., the used implementation and parameter selection of the competitors are not explained.

## Minor comments
  * Typos Line 151: neighbos, line 474: exits
  * There is some mix-up of references, especially when referring to the Appendix, which should be corrected: Figure 6 is in the Appendix and not in the submission, so it should be referenced this way. There is no Figure 5 referenced in the submission (maybe 5 and 6 could be swapped). In Question 3 d) you state that you include the total amount of compute and the type of resources used in Appendix H. They are in Appendix F (maybe they should be moved closer to Appendix H).
  * The presentation of the results and examples could be improved to make them readable in grayscale.
  * PS-VAE is called GP-VAE in Table 1

---

> ### Author Response · Authors · 2022-08-02
> **Part 3**
>
> > **Q4**: More illustration about the relation between principal subgraphs and normal frequent subgraphs.
>
> Sorry for the insufficient details. Basically, principal subgraphs form a special subset of normal frequent subgraphs, satisfying the following property: for a principal subgraph $S$, any other subgraph $S'$ that intersects with $S$ in a certain molecular graph in the dataset satisfies either $S' \subseteq S$ or $c(S') \leq c(S)$. With the notion of principal subgraphs, we can derive the elegant theoretical properties in Theorem 3.2. More importantly,
> it is efficient to extract principal subgraphs by our algorithm, while for normal frequent subgraphs, there is no existing polynomial algorithm for exhausting all normal frequent subgraphs, making the vocabulary extracting time-consuming.
>
>
> > **Q5**: The complexity analysis in the appendix is not sufficient. It appears to assume that functions like GraphToSMILES have constant running time, which needs justification. This function solves the graph canonization problem, for which no polynomial time algorithm is known in general.
>
> Sorry for the lack of strictness. Though no polynomial time algorithm is known for the graph canonization problem in general domain, there does exist a sophisticated polynomial time algorithm called CANGEN [A] for the molecular graph domain thanks to chemical constraints on the graphs. To be more specific, the GraphToSMILES function using [A] has a complexity of $O(n^2 \log n)$, where $n$ denotes the number of atoms in the graph. Also, the SMILESToGraph function has a lower complexity of $O(n)$. Therefore, the complete complexity of Algorithm 1 is $O(NMen^2\log n)$, where $N$ denotes the predefined size of vocabulary, $M$ denotes the number of molecules in the dataset, and $e$ denotes the maximal number of inter-fragment connections in a single molecule.
> Particularly for small molecules, the largest number of atoms $n$ is restricted by the molecular weight, hence we can practically assume these two functions have constant running time. We have added the above details to Appendix E.
>
>
> [A] Weininger, D. et. al SMILES. 2. Algorithm for generation of unique SMILES notation Journal of Chemical Information and Modeling 1989, 29, 97–101
>
>
>
> > **Q6**: Minor comments.
>
> Thank you for these variable comments. We have located and corrected the raised typos including wrong phrases, mistakenly-referenced figures, etc.

---

> > ### Comment · Reviewer_EktL · 2022-08-07
> > **Thanks for the detailed response**
> >
> > Thank you for providing additional experimental results and detailed complexity analysis. This largely alleviates my reservations.

---

> > > ### Author Response · Authors · 2022-08-08
> > > **Thanks for your comments**
> > >
> > > Thanks for your insightful comments, which help improve our paper.

---

> ### Author Response · Authors · 2022-08-02
> **Part 2**
>
> | benchmark                | JTVAE                 | GA                    | PS-VAE (ours)          |
> | ------------------------ | --------------------- | --------------------- | ---------------------- |
> | ZINC250K                 |                       |                       |                        |
> | Celecoxib rediscovery    | 0.245                 | 0.073                 | 0.451                  |
> | Troglitazone rediscovery | 0.171                 | 0.080                 | 0.254                  |
> | Thiothixene rediscovery  | 0.225                 | 0.091                 | 0.286                  |
> | Aripiprazole similarity  | 0.351                 | 0.240                 | 0.432                  |
> | Albuterol similarity     | 0.426                 | 0.437                 | 0.611                  |
> | Mestranol similarity     | 0.278                 | 0.337                 | 0.515                  |
> | C11H24                   | 0.004                 | 0.244                 | 0.634                  |
> | C9H10N2O2PF2Cl           | 0.200                 | 0.659                 | 0.782                  |
> | Median molecules 1       | 0.161                 | 0.190                 | 0.263                  |
> | Median molecules 2       | 0.166                 | 0.111                 | 0.199                  |
> | Osimertinib MPO          | 0.693                 | 0.585                 | 0.800                  |
> | Fexofenadine MPO         | 0.607                 | 0.495                 | 0.715                  |
> | Ranolazine MPO           | 0.330                 | 0.280                 | 0.688                  |
> | Perindopril MPO          | 0.370                 | 0.239                 | 0.429                  |
> | Amlodipine MPO           | 0.442                 | 0.413                 | 0.544                  |
> | Sitagliptin MPO          | 0.160                 | 0.243                 | 0.439                  |
> | Zaleplon MPO             | 0.393                 | 0.259                 | 0.495                  |
> | Valsartan SMARTS         | 0                     | 0                     | 5.398$\times 10^{-12}$ |
> | Deco Hop                 | 0.576                 | 0.530                 | 0.601                  |
> | Scaffold Hop             | 0.440                 | 0.375                 | 0.486                  |
>
> | benchmark                | JTVAE                 | GA                    | PS-VAE (ours)          |
> | ------------------------ | --------------------- | --------------------- | ---------------------- |
> | QM9                      |                       |                       |                        |
> | Celecoxib rediscovery    | 0.056                 | 0.106                 | 0.210                  |
> | Troglitazone rediscovery | 0.070                 | 0.110                 | 0.152                  |
> | Thiothixene rediscovery  | 0.047                 | 0.061                 | 0.146                  |
> | Aripiprazole similarity  | 0.099                 | 0.095                 | 0.214                  |
> | Albuterol similarity     | 0.301                 | 0.243                 | 0.463                  |
> | Mestranol similarity     | 0.170                 | 0.134                 | 0.198                  |
> | C11H24                   | 9.734$\times 10^{-7}$ | 2.148$\times 10^{-7}$ | 0.015                  |
> | C9H10N2O2PF2Cl           | 0.182                 | 0.066                 | 0.381                  |
> | Median molecules 1       | 0.226                 | 0.185                 | 0.261                  |
> | Median molecules 2       | 0.063                 | 0.062                 | 0.112                  |
> | Osimertinib MPO          | 0.408                 | 0.556                 | 0.636                  |
> | Fexofenadine MPO         | 0.217                 | 0.403                 | 0.373                  |
> | Ranolazine MPO           | 0.004                 | 0.170                 | 0.018                  |
> | Perindopril MPO          | 0.104                 | 0.061                 | 0.205                  |
> | Amlodipine MPO           | 0.192                 | 0.166                 | 0.303                  |
> | Sitagliptin MPO          | 2.556$\times 10^{-6}$ | 1.688$\times 10^{-6}$ | 0.319$\times 10^{-3}$  |
> | Zaleplon MPO             | 6.564$\times 10^{-5}$ | 0.03                  | 0.085                  |
> | Valsartan SMARTS         | 0                     | 0                     | 0                      |
> | Deco Hop                 | 0.505                 | 0.525                 | 0.545                  |
> | Scaffold Hop             | 0.341                 | 0.361                 | 0.393                  |

---

> ### Author Response · Authors · 2022-08-02
> **Part 1**
>
> Thank you for your insightful review. We first answer your questions (Q1-Q3) and then provide responses to the raised weaknesses.
>
> > **Q1**: Are the generated molecules chemically valid and how is this guaranteed?
>
> Yes, the generated molecules are chemically valid. Just like previous works (e.g. GCPN, GraphAF), we adopt a valency check during each addition of chemical bonds when connecting generated subgraphs. This means we only add bonds that will not cause violation of valency to the graph to ensure the generated molecules are chemically valid. Besides, the cycle check ensures that only stable rings with low strains will be generated.
>
>
> > **Q2**: Which implementation and parameter selection has been used for the different methods?
>
> For JT-VAE, GCPN, GraphAF, GA, HierVAE, FREED, and MARS, we use their officially released implementations and the default hyperparameters used in their papers in our experiments. For MRNN, since no official implementation is provided, we directly copy the results from its original paper under the same setting as our paper.
>
> > **Q3**: Couly you provide additional results of the different reference methods? 1) FREED and MRNN in Tables 1, 3 and 4. 2) MARS in Tables 1-3. 3) JT-VAE, GCPN, GraphAF, and GA in Table 4.
>
> To address your concern, we have tried our best to add the required results.
>
> （1）FREED is specially designed for protein-binding molecular design, which is practically not suitable for targeting other properties (the experiments in Table 1, 3). Indeed, in Table 3, we have implemented F-VAE\*, a variant of FREED that utilizes its vocabulary but with our decoder to better fit the case of constrained property optimization. As for MRNN, it is hard to be included in other cases besides Table 2, since there is no official implementation source.
>
> (2) We have additionally implemented MARS, with the performance for the experiments in Tables 1-3 below:
>
> | Dataset  | Model         | Uniq  | Novelty | KL Div | FCD   |
> | -------- | ------------- | ----- | ------- | ------ | ----- |
> | ZINC250K | MARS          | 0.737 | 0.737   | 0.798  | 0.271 |
> |          | PS-VAE (ours) | 0.997 | 0.997   | 0.850  | 0.318 |
> | QM9      | MARS          | 0.659 | 0.612   | 0.547  | 0.123 |
> |          | PS-VAE (ours) | 0.673 | 0.523   | 0.921  | 0.659 |
>
> | Model         | PlogP(1st) | PlogP(2nd) | PlogP(3rd) | QED(1st) | QED(2nd) | QED(3rd) |
> | ------------- | ---------- | ---------- | ---------- | -------- | -------- | -------- |
> | MARS          | 7.24       | 6.44       | 6.43       | 0.944    | 0.943    | 0.942    |
> | PS-VAE (ours) | 13.95      | 13.83      | 13.65      | 0.948    | 0.948    | 0.948    |
>
> | Model         | $\delta=0.2$  | $\delta=0.4$  | $\delta=0.6$  |
> | ------------- | ------------- | ------------- | ------------- |
> | MARS          | 4.13$\pm$1.23 | 2.41$\pm$0.76 | 1.21$\pm$0.64 |
> | PS-VAE (ours) | 6.42$\pm$1.86 | 4.19$\pm$1.30 | 2.52$\pm$1.12 |
>
> The results here show our PS-VAE consistently achieves better performance over MARS.
>
> (3) We additionally present the preformance of JTVAE and GA in Table 4 here. We do not report those of GCPN and GraphAF because it is impractical to run them for the guacamol goal-directed benchmarks, as they suffer from poor efficiency in training and finetuning (it needs to be finetuned for each of the 20 properties seperatedly). Clearly, our model achieves the best performance in general.

---

### Official Review · Reviewer_ESyH · 2022-07-11

**Rating:** 6
**Confidence:** 3
**Soundness:** 3 good
**Presentation:** 2 fair
**Contribution:** 3 good

**Summary:**

This paper proposes a molecular generation model based on the vocabulary which consists of frequently observed subgraphs. Unlike existing subgraph-based molecular generation methods using chemical domain knowledge, the vocabulary is constructed in an autoregressive manner based on the dataset. It produces molecular graphs by generating subgraphs that act as building blocks and predicting the edges between them.

**Questions:**

- Although it is forced not to have a ring structure except for a ring with nodes 5 or 6 by the cycle check of Algorithm 3, actual molecular data may have a ring of a different size. Is there a reason to limit the size of cycles?
- What is the advantage of HierVae and FREED in line 298?
- I couldn't find any experimental results supporting the sentence "Otherwise the majority of the benchmarks are easy to be optimized to the upper bound"(line 270) in [42].
- In the proposed method, the decomposed subgraphs are not overlapped. This is different from JT-VAE and HierVAE, which generate subgraphs that can overlap with others and find overlapped parts to connect edges. Is there any advantage to generating non-overlap subgraphs?
- Is there any correlation between the extracted substructures and other properties except PlogP?

**Limitations:**

If there is no significant correlation between the frequently observed subgraph and the desired property, the influence of vocabulary could be diminished.


**Strengths And Weaknesses:**

Strengths

- Unlike using predefined substructures such as rings or functional groups, the proposed approach is possible to adaptively extract subgraphs that frequently appear in datasets. Hence, the proposed model uses a vocabulary that reflects the characteristics of the dataset.

- The properties of extracted subgraphs are well analyzed through the theorem.

- The proposed model has high performance compared to baseline methods.

- Some analysis of extracted vocabulary is shown. The fact that high-correlated fragments to PlogP are generated frequently in the PlogP-optimization task strengthens the proposed method. It would have been better to add experimental results for QED or other properties.

Weaknesses and Questions:

- In the experiment part, HierVAE and H-VAE are used as the baseline method. Is there a reason why these methods are chosen as the baseline? Since HierVAE is a model focused on polymer compound production, it would not be the proper baseline for the ZINC250k or QM9 dataset.

- It is unclear why Theorem 3.2 is useful to model molecules. The connection between the theorem and the experimental results are vague and not clearly explained.

- There is an ambiguity in Algorithm 1. The rightmost molecule in Figure 2 (b) has two C-C subgraphs. If the right C-C bond is merged in iteration 1, all subsequent iterations will be changed. In this case, is there a way to determine which fragments to merge?

- The baseline method used in the experiment is insufficient. MARS or FREED is a recent model, but not used in all experiments. Additional experiments compared with GEGL [1], GraphDF [2], and GraphCNF [3] will help to understand the performance of the presented model.

[1] Ahn, Sungsoo, et al. "Guiding deep molecular optimization with genetic exploration." *Advances in neural information processing systems* 33 (2020): 12008-12021.

[2] Luo, Youzhi, Keqiang Yan, and Shuiwang Ji. "GraphDF: A discrete flow model for molecular graph generation." *International Conference on Machine Learning*. PMLR, 2021.

[3] Lippe, Phillip, and Efstratios Gavves. "Categorical normalizing flows via continuous transformations." *arXiv preprint arXiv:2006.09790* (2020).

Minor Errors

- Typo in line 265 (effacy)
- In 164 lines, it is written that <bos> and <eos> token are used, but in Figure 3, <start> and <end> token is shown.

---

> ### Author Response · Authors · 2022-08-02
> **Part 3**
>
> > **Q8**: In the proposed method, the decomposed subgraphs are not overlapped. This is different from JT-VAE and HierVAE, which generate subgraphs that can overlap with others and find overlapped parts to connect edges. Is there any advantage to generating non-overlap subgraphs?
>
> Thank you for raising this valuable point. We list the advantage of generating non-overlap subgraphs here.
> - Non-overlap subgraphs are more flexible and expressive for representing molecules because the connections between two subgraphs are not strictly restricted to be parts of the subgraphs themselves. For example, suppose we have a vocabulary with only C-C, and we want to merge two C-C. For non-overlap methods, we can connect any types of bonds that conform to the valency rules between them, yielding the possible combinations: C-C=C-C / C-C-C-C / C-C$\equiv$C-C. By contrast, with overlap methods, we can only overlap one carbon on the two C-C to form C-C-C.
> - Another minor problem is that overlap methods are commonly less efficient. For one thing, when generating molecules of the same size, overlap methods usually produce longer subgraph sequences than non-overlap methods do (as in Table 7 in Appendix F). For another, overlap methods need to incorporate enumerations of different overlapping poses before the final decision, while non-overlap methods only need to perform a one-pass prediction of the inter-subgraph edge matrix.
>
>
> > **Q9**: Is there any correlation between the extracted substructures and other properties except PlogP?
>
> Yes, we did draw the correlation in Figure 6 for QED in Appendix. It shows a similar pattern that subgraphs negatively correlated with QED are suppressed and those positively correlated with QED are encouraged in the QED-optimization setting.
>
>
> > **Q10**: limitations. If there is no significant correlation between the frequently observed subgraph and the desired property, the influence of vocabulary could be diminished.
>
> We agree with the reviewer that the influence is related to the correlation between frequently observed subgraphs and the desired property. Fortunately, correlations do exist in our experiments. Also, even if there is no significant correlation between them, the vocabulary can still enhance the model in the distribution learning task.

---

> ### Author Response · Authors · 2022-08-02
> **Part 2**
>
> > **Q4**: The baseline method used in the experiment is insufficient. MARS or FREED is a recent model, but not used in all experiments. Additional experiments compared with GEGL [A], GraphDF [B], and GraphCNF [C] will help to understand the performance of the presented model.
>
> Thank you for your advice. We select GraphDF [B] for comparison as it is the newest method among the raised papers by the reviewer [A,B,C]. We summarize the results in the tables below.
>
> **Results for the distribution-learning benchmarks:**
>
> | Dataset  | Model         | Uniq  | Novelty | KL Div | FCD   |
> | -------- | ------------- | ----- | ------- | ------ | ----- |
> | ZINC250K | GraphDF       | 0.998 | 0.998   | 0.459  | 0.001 |
> |          | PS-VAE (ours) | 0.997 | 0.997   | 0.850  | 0.318 |
> | QM9      | GraphDF       | 0.672 | 0.672   | 0.601  | 0.137 |
> |          | PS-VAE (ours) | 0.673 | 0.523   | 0.921  | 0.659 |
>
> **Results for the optimization task:**
>
> | Model         | PlogP(1st) | PlogP(2nd) | PlogP(3rd) | QED(1st) | QED(2nd) | QED(3rd) |
> | ------------- | ---------- | ---------- | ---------- | -------- | -------- | -------- |
> | GraphDF       | 13.70      | 13.18      | 13.17      | 0.948    | 0.948    | 0.948    |
> | PS-VAE (ours) | 13.95      | 13.83      | 13.65      | 0.948    | 0.948    | 0.948    |
>
> **Results for the constrained optimization task:**
>
> | Model         | $\delta=0.2$  | $\delta=0.4$  | $\delta=0.6$  |
> | ------------- | ------------- | ------------- | ------------- |
> | GraphDF       | 5.62$\pm$1.65 | 4.13$\pm$1.41 | 1.72$\pm$1.15 |
> | PS-VAE (ours) | 6.42$\pm$1.86 | 4.19$\pm$1.30 | 2.52$\pm$1.12 |
>
> In general, our method still outperforms GraphDF in most cases. Although  GraphDF behaves competitively on Uniqueness and Novelty for the distribution-learning benchmarks, it is much worse on KL divergence and FCD, which implies its defected ability in learning the molecular distribution.
>
> Besides, it is kind of impractical to run GraphDF for Guacamol goal-directed benchmarks due to the time and resource limitations, as GraphDF needs to be finetuned for each target property, while our method and other compared baselines only need to train a predictor for all targets at once. This again supports the advantage of our method in efficiency.
>
> [A] Ahn, Sungsoo, et al. "Guiding deep molecular optimization with genetic exploration." *Advances in neural information processing systems* 33 (2020): 12008-12021.
>
> [B] Luo, Youzhi, Keqiang Yan, and Shuiwang Ji. "GraphDF: A discrete flow model for molecular graph generation." *International Conference on Machine Learning*. PMLR, 2021.
>
> [C] Lippe, Phillip, and Efstratios Gavves. "Categorical normalizing flows via continuous transformations." *arXiv preprint arXiv:2006.09790* (2020).
>
> > **Q5**: Although it is forced not to have a ring structure except for a ring with nodes 5 or 6 by the cycle check of Algorithm 3, actual molecular data may have a ring of a different size. Is there a reason to limit the size of cycles?
>
> Yes, we limit the size of cycles due to the chemical knowledge that rings with nodes 5 or 6 are much more stable than others. Since the cycle check is just an ad-hoc operation, it is easy to change the number of nodes to allow rings of different sizes.
>
>
>
> > **Q6**: What is the advantage of HierVae and FREED in line 298?
>
> Sorry for the confusion here. In line 298, we initially mean that the vocabulary constructed by our method shows advantage over HierVAE and FREED because the majority of fragments in their vocabulary are not frequent fragments.
>
>
>
> > **Q7**: I couldn't find any experimental results supporting the sentence "Otherwise the majority of the benchmarks are easy to be optimized to the upper bound"(line 270) in [42].
>
> We apologize for the typo here. The correct reference should be [1]. In Table 2 of [1], many of the properties are optimized to 1.0 or near 1.0 by several baselines pretrained on ChEMBL. This is probably because ChEMBL contains too many molecules and the methods pretrained on it are capable of fitting the downstream tasks easily.

---

> ### Author Response · Authors · 2022-08-02
> **Part 1**
>
> Thank you for your insightful and detailed review!
>
> > **Q1**: Why choose HierVAE or H-VAE as baselines?
>
> Thanks for this comment. We provide the reasons here. In the literature of molecular generation, this method is state-of-the-art and representative by using manual rules to break molecules into subgraph vocabulary, which is a desirable counterpart to verify the effectiveness of our proposed vocabulary construction method. Besides, according to its original paper [19], it was also applicable to small molecules and achieved moderate results.
>
> > **Q2**: Why theorem 3.2 is useful?
>
> We apologize for the lack of the clarity regarding Theorem 3.2.
> It proves the theoretical efficacy of our subgraph extracting algorithm in terms of monotonicity, significance, and completeness. As for monotonicity, it ensures that the subgraphs with higher frequencies are always extracted before those with lower frequencies. This is important because subgraphs with higher frequencies are more likely to reflect the frequent patterns and should be included into the vocabulary earlier. Significance indicates that each extracted subgraph is a principal subgraph that basically represents the “largest” repetitive pattern in size within the data. Table 4 verifies the experimental benefit of extracting principal subgraphs against common subgraphs from manually designed rules or chemical fragment libraries. Completeness means our algorithm is expressive enough to represent (at least contain) any potential principal subgraph.
>
>
> > **Q3**: There is an ambiguity in Algorithm 1. The rightmost molecule in Figure 2 (b) has two C-C subgraphs. If the right C-C bond is merged in iteration 1, all subsequent iterations will be changed. In this case, is there a way to determine which fragments to merge?
>
> Thanks for this sharp observation. In fact, Algorithm 1 can somehow tolerate such ambiguity. By checking the example in Figure 2 (b), the extraction trajectory of the vocabulary is {C} -> {C, CC} -> {C, CC, C=CC}. As suggested by the reviewer, if the right C-C bond is merged in iteration 1, the trajectory will still be {C} -> {C, CC} -> {C, CC, C=CC} because C=CC is still the most frequent fragment after iteration 1. Even if C=CC is not extracted due to the ambiguity here, during the vocabulary extension, C=CC or another principal subgraph containing C=CC will be extracted in the subsequent iterations. This is guaranteed by the completeness of Theorem 3.2 that any principal subgraph (C=CC is a principal subgraph) can be represented (at least contained) by a certain extracted fragment if the size of the vocabulary is sufficiently large. We accept that there could be ambiguity, and we simply resort to the random method during the current extraction process and find it works promisingly in our experiments.

---

### Official Review · Reviewer_B6na · 2022-07-11

**Rating:** 7
**Confidence:** 4
**Soundness:** 4 excellent
**Presentation:** 4 excellent
**Contribution:** 3 good

**Summary:**

The paper proposes a new method for molecular consisting of a principle subgraph extraction step, a fragment prediction step, and an assembling step to combine the fragments and obtain the final generated graph. The major novelty is the extraction of the so-called “principle subgraph” from the dataset. The two-step molecule generation process is based on VAE and relies on the vocabulary of fragments (principle subgraphs) that is built in the first step. Experiments on distribution-learning, (constrained) property optimization, and GuacaMol goal-directed benchmarks show the superiority of the proposed PS-VAE method.

**Questions:**

Please see above.

**Limitations:**

The authors put N/A as the response to the potential societal impact in the last page of the main paper.

**Strengths And Weaknesses:**

Strengths:
1. The proposed method is relatively novel, considering that it is the first work to use frequent subgraphs in the molecule generation domain. Such use of subgraphs is a promising direction and likely to spark future research in this task and other related tasks.
2. In the appendix, careful hyperparameter selection for N, the number of fragments in the vocabulary, is performed using a curve of entropy-sparsity trade-off. Complexity analysis, runtime cost, and additional details on the method, the experiments, and the experimental results are shown too in the appendix.
3. The paper is well-written, with ample examples and illustrations to facilitate the understanding of the various proposed components.

Weaknesses:
1. Since the proposed method consistently outperforms baseline methods, it is important to know which component provides the most useful information. The authors have done a reasonably good job of ablation studies, e.g. by replacing the built vocabulary with other models’ vocabulary. In Section B, the authors talk about the trade-off with the tuning of N, the number of fragments, but it is still unclear what final performance PS-VAE would achieve if some other choices of N are adopted, e.g. N = 100, N = 800, etc. This requires running the whole model multiple times with different choices of N, and would clearly show the performance trend and experimentally validate the claimed trade-off.
2. Although the authors answer “Yes” for whether error bars are shown, it seems only Table 6 contains error bars without further explanation why this is the case.
3. In Section F, the authors compare the running time of proposed method with two baseline models, but it would be better to compare with more baselines or at least show the complexity associated with all the methods that are compared across the three tasks in the main paper. So far the authors claim the proposed method “exhibits higher computational efficiency than certain widely-used baselines”, so this is a technically correct statement, albeit leaving readers wonder about the comparison to other methods adopted in this domain.

---

> ### Author Response · Authors · 2022-08-02
> **Thank you for your review**
>
> Thank you for your insightful review and positive comments!
>
> > **Q1**: In Section B, the authors talk about the trade-off with the tuning of N, the number of fragments, but it is still unclear what final performance PS-VAE would achieve if some other choices of N are adopted, e.g. N = 100, N = 800, etc. This requires running the whole model multiple times with different choices of N, and would clearly show the performance trend and experimentally validate the claimed trade-off.
>
> Nice suggestion! We have additionally conducted experiments on (constrained) optimization tasks with N=100, 300, 500, and 700. The results are provided in the following tables.
>
> **Results for the optimization task:**
>
> | N    | PlogP(1st) | PlogP(2nd) | PlogP(3rd) | QED(1st) | QED(2nd) | QED(3rd) |
> | ---- | ---------- | ---------- | ---------- | -------- | -------- | -------- |
> | 100  | 10.30      | 10.06      | 9.96       | 0.948    | 0.948    | 0.948    |
> | 300  | 13.95      | 13.83      | 13.65      | 0.948    | 0.948    | 0.948    |
> | 500  | 12.57      | 12.24      | 12.21      | 0.948    | 0.948    | 0.948    |
> | 700  | 8.41       | 8.20       | 8.10       | 0.948    | 0.948    | 0.948    |
>
> **Results for the contrained optimization task:**
>
> | N    | $\delta=0.2$  | $\delta=0.4$  | $\delta=0.6$  |
> | ---- | ------------- | ------------- | ------------- |
> | 100  | 5.17$\pm$1.66 | 3.65$\pm$1.30 | 2.30$\pm$1.04 |
> | 300  | 5.42$\pm$2.30 | 3.70$\pm$1.54 | 2.31$\pm$1.12 |
> | 500  | 6.42$\pm$1.86 | 4.19$\pm$1.30 | 2.52$\pm$1.12 |
> | 700  | 4.93$\pm$1.81 | 3.61$\pm$1.37 | 2.26$\pm$1.13 |
>
> We observe that the best values of N for the above two tasks are 300 and 500, respectively, both of which are consistent with the optimal points by the trade-off curves in Figure 5 of Section B. It suggests that in practice we can tune the value of N by the method proposed in Appendix B. All above results are contained in Appendix B.
>
> > **Q2**: It seems only Table 6 contains error bars without further explanation why this is the case.
>
>
> Thank you for raising this valuable point! We provide the explanation here. For guacamol benchmarks [7], they calculate metrics with respect to the global data distribution, making the results insensitive to randomness. For the optimization tasks, they simulate the real scenario where a bunch of candidates are produced (about 10,000 candidates) and the high-scoring ones (e.g. top-3) are selected for further development, which means the highest scores achieved by each model are relatively stable. For the constrained optimization tasks (Table 6), we report the error bars to evaluate the robustness of the model, since different input molecules may result in different improvements. We have revised the checklist answer accordingly.
>
>
> > **Q3**: In Section F, the authors compare the running time of proposed method with two baseline models, but it would be better to compare with more baselines or at least show the complexity associated with all the methods that are compared across the three tasks in the main paper.
>
> Again, nice advice! We further compare the running time of several generative baselines and present the results in the table below. The highest computational efficiency of our method is still observed. We do not include the methods like GA and MARS, since they adopt the training-on-the-fly strategy and it is unfair to perform comparisons with them. The following table is added to Appendix F.
>
> | Model         | Training   | Inference | Avg Step |
> | ------------- | ---------- | --------- | -------- |
> | JT-VAE        | 24 hours   | 20 hours  | 15.50    |
> | GCPN          | 14 hours   | 20 hours  | 38.21    |
> | GraphAF       | 7 hours    | 10 hours  | 56.88    |
> | HierVAE       | 10.9 hours | 1.2 hours | 36.90    |
> | PS-VAE (ours) | 1.2 hours  | 0.3 hours | 6.84     |

---

### Author Response · Authors · 2022-08-02
**Summary**

We sincerely thank all reviewers for their valuable and constructed comments. We have addressed the following main concerns and revised our paper accordingly:

1. We have updated MARS in Table 1-3 and JTVAE and GA in Table 4.
2. We have additionally measured the runtime cost of GCPN and HierVAE and added them in Appendix F.
3. We have provided results on (constrained) property optimization of our PS-VAE with different vocabulary sizes in Appendix B for a more direct illustration of the effect of granularity.
4. We have corrected some typos and ambiguity in our presentation mentioned by the reviewers to avoid misunderstandings.

---

### Meta-Review · Area_Chair_Q3DX · 2022-08-22

**Recommendation:** Accept
**Confidence:** Certain

**Metareview:**

This paper proposes a molecule generation method using frequent subgraphs. There was a positive consensus among reviewers that paper is novel (B6na, EktL) and well-analyzed (B6na, ESyH), and minor suggestions for improved evaluations and presentation (EktL, ESyH) were well-handled during rebuttals to alleviate reviewer concerns.

**Award:**

No

---

### Decision · Program_Chairs · 2022-09-14

Accept